# Advantage Weighted Matching:
# Aligning RL with Pretraining in Diffusion Models

**Shuchen Xue** [1 2]   **Chongjian Ge** [2]   **Shilong Zhang** [2 3]   **Yichen Li** [2 4]   **Zhi-Ming Ma** [1]

## Abstract

Reinforcement Learning (RL) has emerged as a central paradigm for advancing Large Language Models (LLMs), where both pre-training and RL post-training stages are grounded in the same log-likelihood formulation. In contrast, recent RL approaches for diffusion models, most notably Denoising Diffusion Policy Optimization (DDPO), optimize an objective different from the pretraining objectives–score/flow matching loss. In this work, we establish a novel theoretical analysis: DDPO is an implicit form of score/flow matching with noisy targets, which increases variance and slows convergence. Building on this analysis, we introduce **Advantage Weighted Matching (AWM)**, a policy-gradient method for diffusion. It uses the score/flow-matching loss and reweights each sample by its advantage. In effect, AWM raises the influence of high-reward samples and suppresses low-reward ones while keeping the modeling objective identical to pretraining. This simple yet effective design yields substantial benefits: on the GenEval, OCR, and PickScore benchmarks, AWM delivers up to a **34× speedup** over Flow-GRPO (which builds on DDPO), when applied to Stable Diffusion 3.5 Medium and FLUX, without compromising generation quality. Code is available at https://github.com/scxue/advantage_weighted_matching.

## 1. Introduction

Diffusion Models (Sohl-Dickstein et al., 2015; Ho et al., 2020; Song et al., 2020) and their variants, (Gaussian) Flow Matching (Peluchetti, 2023; Lipman et al., 2022; Albergo et al., 2023) are de facto generation paradigms for continu-

---

[1]University of Chinese Academy of Sciences [2]Adobe Research [3]The University of Hong Kong [4]Massachusetts Institute of Technology. Correspondence to: Chongjian Ge <cge@adobe.com>.

*Proceedings of the 43rd International Conference on Machine Learning*, Seoul, South Korea. PMLR 306, 2026. Copyright 2026 by the author(s).

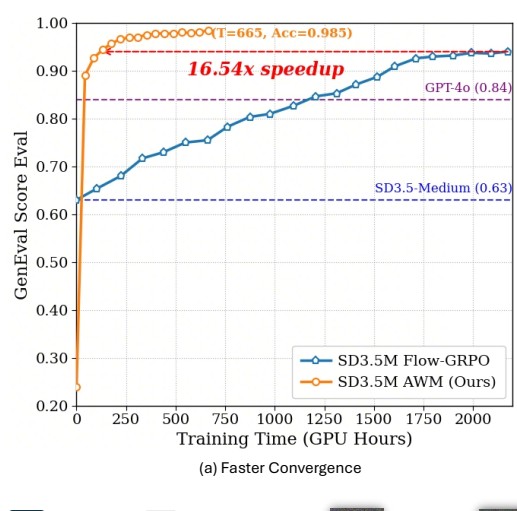

(a) Faster Convergence

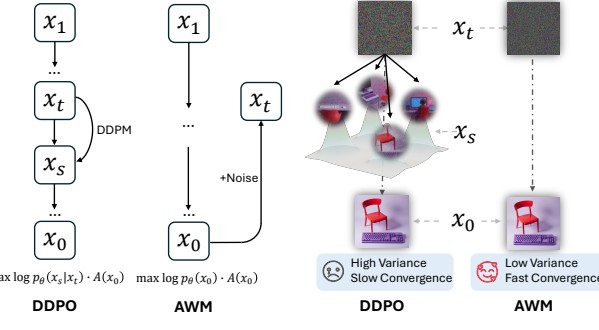

(b) Formulations: DDPO vs. AWM       (c) Noisy vs. clean DSM variance

*Figure 1.* **AWM reduces variance and speeds up RL.** (a) *Convergence:* On GenEval, AWM reaches the same quality with 8× fewer GPU hours than Flow-GRPO. (b) *Formulations:* DDPO optimizes per-step Gaussian likelihood on $x_{t-1}$, whereas AWM applies advantage-weighted score/flow matching on $x_0$. (c) *Target variance:* DSM with noisy target $x_s$ (DDPO implicitly doing) has higher variance than DSM with clean $x_0$ (AWM).

ous domains like image and video synthesis (Nichol et al., 2021; Saharia et al., 2022; Ramesh et al., 2022; Rombach et al., 2022; Chen et al., 2023b; Brooks et al., 2024). Motivated by the success of Reinforcement Learning (RL) for Large Language Models (LLMs) (Ziegler et al., 2019; Stiennon et al., 2020; Ouyang et al., 2022; Bai et al., 2022; Guo et al., 2025), there has been a growing interest in extending RL methodologies to diffusion models.

A key difference between RL algorithms used in LLMs and diffusion lies in the alignment of training objectives. In the

autoregressive setting (*e.g.* LLMs), both pre-training and RL post-training ultimately optimize the same log-likelihood objective with different weights. In contrast, diffusion pre-training relies on score/flow matching losses that are closely related to maximum likelihood through the Evidence Lower Bound (ELBO) (Song et al., 2021) via forward process, while recent RL post-training methods for diffusion have largely adopted Denoising Diffusion Policy Optimization (DDPO) (Black et al., 2023), including Flow-GRPO (Liu et al., 2025a) and Dance-GRPO (Xue et al., 2025). As shown in Figure 1(b) (left), these approaches formulate de-noising as a multi-step MDP, where each reverse-time step is an action, and the one-step reverse transition $p_{\boldsymbol{\theta}}(\boldsymbol{x}_{t-1}|\boldsymbol{x}_t, \boldsymbol{c})$ acts as the policy. Under standard DDPM samplers, this transition is isotropic Gaussian, enabling tractable per-step log-likelihoods and supporting policy-gradient methods, e.g., REINFORCE (Williams, 1992).

Although effective, DDPO-based RL post-training leads to an optimization objective that diverges from the score or flow matching objectives used during pretraining. The discrepancy raises fundamental questions: ***Why do diffusion RL post-training and pretraining employ different likelihood formulations? Consequently, what does the DDPO loss objective actually optimize?*** To answer this question, we establish a theoretical connection in Section 2: DDPO is implicitly doing denoising score matching (DSM) with noisy data target. We prove that using noisy data in-creases the variance of the DSM target compared to the clean-data DSM objectives used in pretraining (shown in Figure 1(c) (right)). These theoretical insights are validated in Section 2 through pretraining experiments on CIFAR-10 and ImageNet-64, where we show that noisy-DSM objec-tives align with the DDPO target but converge more slowly, consistent with our variance analysis.

We propose Advantage Weighted Matching (AWM), a sim-ple yet effective objective that directly incorporates reward signals into score/flow matching without relying on reverse-time discretization. In contrast to DDPO, which depends on per-step likelihoods derived from reverse-time discretiza-tion, AWM uses the same score/flow-matching loss as diffu-sion pretraining while reweighting each sample by its advan-tage, as shown in Figure 1(b) (right). Intuitively, AWM am-plifies high-reward samples and downweights low-reward ones, while preserving the original pretraining objective. This design offers several benefits. First, it avoids the vari-ance amplification caused by noisy data target, leading to faster convergence. Second, it decouples training from sam-pling: AWM supports any sampler, unlike DDPO methods tied to Euler–Maruyama discretization. Finally, by ground-ing RL post-training in the same score/flow matching frame-work as pretraining, AWM restores conceptual symmetry between diffusion models and LLMs, where both stages op-timize the same objective with reward-dependent weighting.

AWM translates this variance reduction into substantial wall-clock savings without compromising quality. On Stable Diffusion 3.5 Medium (SD3.5M) (Esser et al., 2024), it surpasses Flow-GRPO's (Liu et al., 2025a) GenEval score (0.98 vs. 0.95) with a 16.54× speed-up (Figure 1(a)). It achieves comparable OCR accuracy with 19.13× less com-pute and the PickScore with 34.67× less compute. On FLUX (Labs, 2024), AWM shows 8.5× (OCR) and 6.8× (PickScore) speedups (Figure 5 a–d), reaching 0.986 OCR accuracy. Our contributions are threefold:

- **DDPO–DSM Equivalence.** We prove that, up to dis-cretization error, maximizing DDPO's per-step Gaussian likelihood is equivalent to minimizing denoising score matching (DSM) with *noisy data* at the same time step, regardless of parameterization (score or velocity).

- **Additional Variance for DDPO from Noisy targets.** We show that targeting on noisy data yields a higher-variance estimator of score function than targeting on clean data; we quantify the increase and validate it via controlled pretraining on CIFAR-10 and ImageNet-64, where noisy-DSM converges more slowly under identical settings.

- **Advantage Weighted Matching (AWM).** We introduce Advantage Weighted Matching, a policy gradient method for diffusion with forward process that decouples training from sampling, and re-aligns RL post-training with the pretraining objective. On GenEval, OCR, ImageReward, and PickScore, AWM achieves up to $34\times$ faster training than Flow-GRPO (Liu et al., 2025a) on SD3.5M and FLUX without generation quality degradation.

## 2. DDPO Is Secretly Doing Score Matching

### 2.1. Background on Diffusion and Flow Matching

Diffusion Models (Sohl-Dickstein et al., 2015; Ho et al., 2020; Song et al., 2020) diffuse clean data $\boldsymbol{x}_0 \sim p_{\text{data}}$ from data distribution to noisy data $\boldsymbol{x}_t = \alpha_t \boldsymbol{x}_0 + \sigma_t \boldsymbol{\epsilon}$, where $t \in [0, T]$, $\boldsymbol{\epsilon} \sim \mathcal{N}(\boldsymbol{0}, \boldsymbol{I})$ follows standard Gaussian. $\boldsymbol{x}_T$ exactly or approximately follows a Gaussian distribution. Typically, diffusion models train a noise prediction network $\boldsymbol{\epsilon}_{\boldsymbol{\theta}}$ using $\mathbb{E}_{\boldsymbol{x}_0, \boldsymbol{\epsilon}, t}[\|\boldsymbol{\epsilon}_{\boldsymbol{\theta}}(\boldsymbol{x}_t, t) - \boldsymbol{\epsilon}\|^2]$, which is equivalent to denoising score matching loss up to a time-dependent weight (Vincent, 2011; Song et al., 2020).

Flow Matching (Peluchetti, 2023; Liu et al., 2022; Lip-man et al., 2022; Albergo et al., 2023) with Gaussian priors considers a linear interpolation noising process by defining $\boldsymbol{x}_t = (1 - t)\boldsymbol{x}_0 + t\boldsymbol{\epsilon}$. The flow match-ing models train a velocity prediction network $\boldsymbol{v}_{\boldsymbol{\theta}}$ using $\mathbb{E}_{\boldsymbol{x}_0, \boldsymbol{\epsilon}, t}[\|\boldsymbol{v}_{\boldsymbol{\theta}}(\boldsymbol{x}_t, t) - (\boldsymbol{\epsilon} - \boldsymbol{x}_0)\|^2]$. To simplify the notation, in the remaining part of the paper, we choose the flow match-ing noise schedule by default. Throughout this paper, we use the notations for v-prediction $\boldsymbol{v}_{\boldsymbol{\theta}}$ and score prediction $\boldsymbol{s}_{\boldsymbol{\theta}}$ interchangeably, as they are equivalent parameteriza-

tions and can be converted via a simple algebraic relation: $s_{\boldsymbol{\theta}}(\boldsymbol{x}_t, t) = -\frac{(1-t)\,\boldsymbol{v}_{\boldsymbol{\theta}}(\boldsymbol{x}_t, t)+\boldsymbol{x}_t}{t}$.

The sampling process of diffusion or flow matching models involves solving the probability flow ODE (PF-ODE) (Song et al., 2020): $\mathrm{d}\boldsymbol{x}_t = \boldsymbol{v}_{\boldsymbol{\theta}}(\boldsymbol{x}_t, t)\mathrm{d}t$, or diffusion SDE (Song et al., 2020), where $\boldsymbol{w}_t$ represents the Wiener process[1]:

$$\mathrm{d}\boldsymbol{x}_t = \left[\boldsymbol{v}_{\boldsymbol{\theta}}(\boldsymbol{x}_t, t) + \frac{(\boldsymbol{x}_t + (1-t)\boldsymbol{v}_{\boldsymbol{\theta}}(\boldsymbol{x}_t, t))}{1-t}\right]\mathrm{d}t + \sqrt{\frac{2t}{1-t}}\mathrm{d}\boldsymbol{w}_t, \quad (1)$$

The likelihood $\log p(\boldsymbol{x})$ of data $\boldsymbol{x}$ that diffusion models assign can be exactly computed through instantaneous change of variables as in Neural ODE (Chen et al., 2018) or approximated by an Evidence Lower Bound (ELBO) (Song et al., 2021; Kingma et al., 2021; Kingma & Gao, 2023):

$$-\log p(\boldsymbol{x}) \leq \mathrm{ELBO}(\boldsymbol{x})$$
$$:= \mathbb{E}_{\substack{t \sim \mathcal{U}(0,1) \\ \boldsymbol{\epsilon} \sim \mathcal{N}(\mathbf{0}, \mathbf{I})}} \left[\frac{1-t}{t} \|\boldsymbol{v}_{\boldsymbol{\theta}}(\boldsymbol{x}_t, t) - (\boldsymbol{\epsilon} - \boldsymbol{x}_0)\|^2\right]. \quad (2)$$

The gap between the exact likelihood and ELBO is relatively small (Song et al., 2021).

## 2.2. Denoising Diffusion Policy Optimization (DDPO)

Denoising Diffusion Policy Optimization (DDPO) (Black et al., 2023) casts denoising as a multi-step MDP: the state at step $t$ is $\boldsymbol{x}_t$, the policy is the one-step reverse transition $p_{\boldsymbol{\theta}}(\boldsymbol{x}_{t-1}|\boldsymbol{x}_t)$. Under standard DDPM samplers (or Euler–Maruyama discretization of the reverse-time diffusion SDE), this transition is isotropic Gaussian, which makes the per-step log-likelihood tractable. Concretely, DDPO implements the Euler–Maruyama update of the reverse SDE from Equation (1)[2]:

$$\boldsymbol{x}_{t-\Delta t} = \boldsymbol{x}_t - \left[2\boldsymbol{v}_{\boldsymbol{\theta}}(\boldsymbol{x}_t, t) + \frac{\boldsymbol{x}_t}{1-t}\right]\Delta t + \sqrt{\frac{2t}{1-t}}\sqrt{\Delta t}\boldsymbol{\epsilon}, \quad (3)$$

This implies a Gaussian policy $p_{\boldsymbol{\theta}}(\boldsymbol{x}_{t-\Delta t}|\boldsymbol{x}_t)$ with tractable mean and covariance, yielding the per-step log-likelihood $\log p_{\boldsymbol{\theta}}(\boldsymbol{x}_{t-\Delta t}|\boldsymbol{x}_t)$:

$$-\frac{\left\|\boldsymbol{x}_{t-\Delta t} - \left[\boldsymbol{x}_t - \left(2\boldsymbol{v}_{\boldsymbol{\theta}}(\boldsymbol{x}_t, t) + \frac{\boldsymbol{x}_t}{1-t}\right)\Delta t\right]\right\|^2}{\frac{4t}{1-t}\Delta t} + \mathrm{const.} \quad (4)$$

Then the DDPO's objective is

$$\mathbb{E}_{p_{\boldsymbol{\theta}_{\mathrm{old}}}}\left[\sum_{t=0}^{T} \frac{p_{\boldsymbol{\theta}}(\boldsymbol{x}_{t-1}|\boldsymbol{x}_t, \boldsymbol{c})}{p_{\boldsymbol{\theta}_{\mathrm{old}}}(\boldsymbol{x}_{t-1}|\boldsymbol{x}_t, \boldsymbol{c})} (r(\boldsymbol{x}_0, \boldsymbol{c}) - b(\boldsymbol{c}))\right]. \quad (5)$$

where $b(\boldsymbol{c})$ denotes an arbitrary baseline independent of $\boldsymbol{x}_0$. This baseline does not change the expected gradient but can significantly reduce its variance (Greensmith et al., 2004). The baseline can be computed in various ways (e.g., PPO (Schulman et al., 2017) or GRPO (Shao et al., 2024)) and does not affect the correctness of the objective.

In the on-policy case, the old policy is a stop-gradient copy of the current policy. Let $\mathrm{sg}[\cdot]$ denote the stop-gradient operator. Setting $p_{\boldsymbol{\theta}_{\mathrm{old}}} = \mathrm{sg}[p_{\boldsymbol{\theta}}]$ gives the surrogate

$$\mathbb{E}_{\mathrm{sg}[p_{\boldsymbol{\theta}}]}\left[\sum_{t=0}^{T} \frac{p_{\boldsymbol{\theta}}(\boldsymbol{x}_{t-1}|\boldsymbol{x}_t, \boldsymbol{c})}{\mathrm{sg}[p_{\boldsymbol{\theta}}(\boldsymbol{x}_{t-1}|\boldsymbol{x}_t, \boldsymbol{c})]} (r(\boldsymbol{x}_0, \boldsymbol{c}) - b(\boldsymbol{c}))\right]. \quad (6)$$

Although the likelihood ratio has value one in the forward pass, the denominator is detached from the gradient. Therefore,

$$\nabla_{\boldsymbol{\theta}} \frac{p_{\boldsymbol{\theta}}(\boldsymbol{x}_{t-1}|\boldsymbol{x}_t, \boldsymbol{c})}{\mathrm{sg}[p_{\boldsymbol{\theta}}(\boldsymbol{x}_{t-1}|\boldsymbol{x}_t, \boldsymbol{c})]} = \nabla_{\boldsymbol{\theta}} \log p_{\boldsymbol{\theta}}(\boldsymbol{x}_{t-1}|\boldsymbol{x}_t, \boldsymbol{c}), \quad (7)$$

at the current parameter value. Thus, the on-policy DDPO surrogate induces the same policy-gradient update as optimizing the advantage-weighted per-step log-likelihood:

$$\mathbb{E}_{\mathrm{sg}[p_{\boldsymbol{\theta}}]}\left[\sum_{t=0}^{T} \log p_{\boldsymbol{\theta}}(\boldsymbol{x}_{t-1}|\boldsymbol{x}_t, \boldsymbol{c}) (r(\boldsymbol{x}_0, \boldsymbol{c}) - b(\boldsymbol{c}))\right]. \quad (8)$$

For completeness, we include a detailed description of DDPO in Appendix D.

## 2.3. Denoising Score Matching with Noisy Data

Standard diffusion pretraining optimizes a score/flow-matching objective defined entirely on the *forward process*, using pairs $(\boldsymbol{x}_0, \boldsymbol{x}_t)$:

$$\|\boldsymbol{v}_{\boldsymbol{\theta}}(\boldsymbol{x}_t, t) - (\boldsymbol{\epsilon} - \boldsymbol{x}_0)\|^2. \quad (9)$$

In contrast, DDPO (Equation (3)) formulates denoising as a multi-step Markov decision process on the *reverse process*, where optimization is performed on the per-step reverse transition likelihood $p_{\theta}(\boldsymbol{x}_{t-\Delta t}|\boldsymbol{x}_t)$. Consequently, the two objectives are defined on different data pairs: diffusion pretraining operates on forward-process pairs $(\boldsymbol{x}_0, \boldsymbol{x}_t)$ with clean target, whereas DDPO operates on backward-process pairs $(\boldsymbol{x}_{t-\Delta t}, \boldsymbol{x}_t)$ with noisy target.

Although our analysis is motivated by RL post-training, policy-gradient methods are fundamentally likelihood-based: the reward or advantage weights the gradient of a log-likelihood term. Therefore, before analyzing the full advantage-weighted DDPO update, we first ask what objective is induced by the DDPO likelihood term itself. Concretely, we study the unit-weight per-step likelihood component, which can be viewed as the maximum-likelihood

---

[1]Or arbitrary noise levels (Karras et al., 2022; Xue et al., 2023)

[2]For simplicity, our analysis uses the vanilla reverse-time diffusion SDE. The original DDPO (Black et al., 2023) uses DDPM, which can be viewed as Euler–Maruyama with an exponential integrator (Xue et al., 2023). Flow-GRPO (Liu et al., 2025a) and Dance-GRPO (Xue et al., 2025) use the plain Euler–Maruyama discretization. The discrepancy vanishes as the discretization step $\Delta t \to 0$.

counterpart of the DDPO surrogate. This isolates the modeling objective induced by the reverse-transition likelihood and allows us to compare it directly with the score/flow-matching objectives used in diffusion pretraining. After establishing its equivalence to noisy-target DSM, we extend the argument to the full advantage-weighted DDPO update.

At first glance, these objectives appear fundamentally incompatible—not only because one is likelihood-derived and the other matching-derived, but also because they are defined on different processes and different data pairs. However, this apparent discrepancy disappears once both objectives are analyzed at the level of their underlying joint distributions.

By classical results on time reversal of diffusions (Haussmann & Pardoux, 1986), the joint distribution of $(\boldsymbol{x}_{t-\Delta t}, \boldsymbol{x}_t)$ is identical under the forward and reverse diffusion processes. As a consequence, although the pair $(\boldsymbol{x}_{t-\Delta t}, \boldsymbol{x}_t)$ is generated by the reverse process in DDPO, it can be analyzed equivalently under the forward process. This allows us to reinterpret the DDPO per-step likelihood—originally derived from a discretization of the reverse-time SDE—as a population objective over forward-process samples $(\boldsymbol{x}_{t-\Delta t}, \boldsymbol{x}_t)$.

In the following analysis, we consider an idealized (oracle) DDPO setting: we focus on a single reverse step at time $t$, analyze *population-level* objectives, and omit discretization error induced by Euler-Maruyama. Under these, we show that maximizing the DDPO per-step Gaussian likelihood is equivalent to minimizing a denoising score matching objective on $\boldsymbol{x}_t$ with noisy target $\boldsymbol{x}_{t-\Delta t}$. This equivalence serves as the starting point of our theoretical analysis.

**Theorem 2.1** (DDPO Per-Step Likelihood Is Secretly Doing Denoising Score Matching with Noisy Data). *Consider a DDPO objective at a fixed time $t$, where we analyze population-level objectives. Let $(\boldsymbol{x}_{t-\Delta t}, \boldsymbol{x}_t) \sim p_{t-\Delta t, t}$ denote their common joint distribution, which is identical under the forward and reverse diffusions (Haussmann & Pardoux, 1986).*

*Then, maximizing the DDPO per-step log-likelihood*

$$\mathbb{E}_{(\boldsymbol{x}_{t-\Delta t}, \boldsymbol{x}_t)} [\log p_{\boldsymbol{\theta}}(\boldsymbol{x}_{t-\Delta t} | \boldsymbol{x}_t)]$$

*induces the same population objective, in the sense that it shares the same set of population minimizers, as minimizing the denoising score matching loss with noisy target:*

$$\mathbb{E}_{(\boldsymbol{x}_{t-\Delta t}, \boldsymbol{x}_t)} \left[ \|\boldsymbol{s}_{\boldsymbol{\theta}}(\boldsymbol{x}_t, t) - \nabla_{\boldsymbol{x}_t} \log p(\boldsymbol{x}_t | \boldsymbol{x}_{t-\Delta t})\|^2 \right].$$

**Remark** Theorem 2.1 characterizes the population-level objective induced by the DDPO per-step likelihood under the exact joint distribution $p_{t-\Delta t, t}$. In practice, DDPO relies on a discretized reverse-time transition (e.g., Euler–Maruyama or DDPM), which introduces additional bias

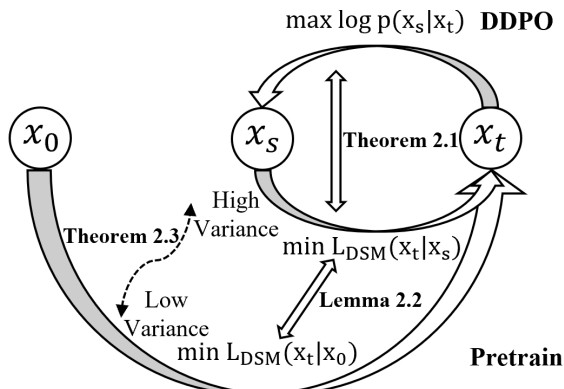

*Figure 2.* **Overview of our analysis**: DDPO is equivalent to denoising score matching with noisy target (Theorem 2.1); although the population minimizer is unchanged (Lemma 2.2), this noisy target introduces higher variance (Theorem 2.3), leading to slower optimization.

when $(\boldsymbol{x}_{t-\Delta t}, \boldsymbol{x}_t)$ is generated from the reverse process. This discretization effect is inherent to the DDPO formulation and is not an approximation introduced by our analysis. By contrast, the forward diffusion process admits an exact transition and does not incur such discretization bias.

The proof can be found in Appendix C. As summarized in Figure 2, Theorem 2.1 establishes that, the DDPO per-step log-likelihood induces a denoising score matching objective defined on forward-process pairs $(\boldsymbol{x}_{t-\Delta t}, \boldsymbol{x}_t)$. At this point, the DDPO objective has been re-expressed as a noisy DSM objective on the forward process. Standard diffusion pretraining optimizes DSM using clean-data pairs $(\boldsymbol{x}_0, \boldsymbol{x}_t)$. The following lemma bridges the two objectives by showing, in expectation, DSM with a noisy target shares the same population minimizer as DSM with clean data.

**Lemma 2.2** (Denoising Score Matching with Noisy Data). *For any time steps $s$ and $t$ such that $0 \le s < t$, optimizing the Denoising Score Matching objective using the noisy data $\boldsymbol{x}_s$*

$$\mathbb{E}_{\boldsymbol{x}_s} \mathbb{E}_{\boldsymbol{x}_t | \boldsymbol{x}_s} \left[ \|\boldsymbol{s}_{\boldsymbol{\theta}}(\boldsymbol{x}_t, t) - \nabla \log p(\boldsymbol{x}_t | \boldsymbol{x}_s)\|^2 \right],$$

*is equivalent to optimizing the standard Score Matching objective:*

$$\mathbb{E}_{\boldsymbol{x}_t} \left[ \|\boldsymbol{s}_{\boldsymbol{\theta}}(\boldsymbol{x}_t, t) - \nabla \log p(\boldsymbol{x}_t)\|^2 \right].$$

*This lemma generalizes the well-known result that Denoising Score Matching with clean data is equivalent to Score Matching (Vincent, 2011), which corresponds to the special case in this lemma when $s = 0$.*

The proof is in Appendix C. Together with Lemma 2.2, the DDPO update can be viewed as optimizing a denoising score matching objective at the same time $t$, which shares the same population minimizer as standard score matching.

**Advantage-weighted implication.** The above two-step connection extends to the full advantage-weighted DDPO surrogate in Equation (5). Let $A(\boldsymbol{x}_0, \boldsymbol{c}) = r(\boldsymbol{x}_0, \boldsymbol{c}) - b(\boldsymbol{c})$, fix the rollout policy $\bar{\boldsymbol{\theta}} = \mathrm{sg}(\boldsymbol{\theta})$, and let $J_t^{\mathrm{DDPO}}$ denote the $t$-th summand of Equation (5). No independence between $A(\boldsymbol{x}_0, \boldsymbol{c})$ and $\boldsymbol{x}_t$ is required: under the rollout joint law, we condition on $(\boldsymbol{x}_0, \boldsymbol{c})$, so the advantage is fixed inside the conditional expectation. Applying Theorem 2.1 and Lemma 2.2 conditionally yields the local-gradient relation

$$
\begin{aligned}
&\nabla_{\boldsymbol{\theta}} J_t^{\mathrm{DDPO}}(\boldsymbol{\theta}; \bar{\boldsymbol{\theta}})\big|_{\boldsymbol{\theta}=\bar{\boldsymbol{\theta}}} \\
&= -\alpha_t \, \nabla_{\boldsymbol{\theta}} \mathbb{E}_{\bar{q}_t} \left[ A(\boldsymbol{x}_0, \boldsymbol{c}) \ell_{\boldsymbol{\theta}, t}^{\mathrm{clean}}(\boldsymbol{x}_0, \boldsymbol{x}_t, \boldsymbol{c}) \right]\big|_{\boldsymbol{\theta}=\bar{\boldsymbol{\theta}}},
\end{aligned}
\tag{10}
$$

where $\alpha_t > 0$, $\bar{q}_t(\boldsymbol{c}, \boldsymbol{x}_0, \boldsymbol{x}_t) = p(\boldsymbol{c}) \pi_{\bar{\boldsymbol{\theta}}}(\boldsymbol{x}_0|\boldsymbol{c}) p(\boldsymbol{x}_t|\boldsymbol{x}_0)$, and $\ell_{\boldsymbol{\theta}, t}^{\mathrm{clean}}(\boldsymbol{x}_0, \boldsymbol{x}_t, \boldsymbol{c}) = \|\boldsymbol{s}_{\boldsymbol{\theta}}(\boldsymbol{x}_t, t, \boldsymbol{c}) - \nabla_{\boldsymbol{x}_t} \log p(\boldsymbol{x}_t|\boldsymbol{x}_0)\|^2$. Here $\pi_{\bar{\boldsymbol{\theta}}}(\boldsymbol{x}_0|\boldsymbol{c})$ is the terminal sample distribution induced by the rollout policy. The full derivation is in Appendix C.3. Thus, locally at the frozen rollout policy, DDPO gradient ascent is equivalent, up to a positive timestep-dependent rescaling, to gradient descent on a signed advantage-weighted clean-target DSM objective. This directly motivates AWM.

What remains is to understand how replacing the clean target $\boldsymbol{x}_0$ with a noisy target $\boldsymbol{x}_s$ affects the variance of the DSM objective. The following theorem formalizes this effect.

**Theorem 2.3** (Variance of Denoising Score Matching with Noisy Data is Larger). *For the two conditional scores*

$$
\nabla_{\boldsymbol{x}_t} \log p(\boldsymbol{x}_t|\boldsymbol{x}_s), \qquad \nabla_{\boldsymbol{x}_t} \log p(\boldsymbol{x}_t|\boldsymbol{x}_0),
$$

*and for any $\boldsymbol{x}_t$ and any $s \in [0, t)$, each is unbiased estimator of $\nabla_{\boldsymbol{x}_t} \log p(\boldsymbol{x}_t)$:*

$$
\begin{aligned}
\mathbb{E}[\nabla_{\boldsymbol{x}_t} \log p(\boldsymbol{x}_t|\boldsymbol{x}_s) \mid \boldsymbol{x}_t] &= \nabla_{\boldsymbol{x}_t} \log p(\boldsymbol{x}_t), \\
\mathbb{E}[\nabla_{\boldsymbol{x}_t} \log p(\boldsymbol{x}_t|\boldsymbol{x}_0) \mid \boldsymbol{x}_t] &= \nabla_{\boldsymbol{x}_t} \log p(\boldsymbol{x}_t).
\end{aligned}
$$

*Moreover, their conditional covariances satisfy*
$$
\begin{aligned}
&\mathrm{Cov}\left(\nabla_{\boldsymbol{x}_t} \log p(\boldsymbol{x}_t|\boldsymbol{x}_s)|\boldsymbol{x}_t\right) \\
&= \mathrm{Cov}\left(\nabla_{\boldsymbol{x}_t} \log p(\boldsymbol{x}_t|\boldsymbol{x}_0)|\boldsymbol{x}_t\right) + \kappa(s, t)\, I \\
&\succeq \mathrm{Cov}\left(\nabla_{\boldsymbol{x}_t} \log p(\boldsymbol{x}_t|\boldsymbol{x}_0)|\boldsymbol{x}_t\right),
\end{aligned}
$$

*where*
$$
\kappa(s, t) = \frac{(1-t)^2\, s^2}{t^2\left(t^2(1-s)^2 - s^2(1-t)^2\right)}.
$$

*In particular,* $\mathrm{Tr}\,\mathrm{Cov}(\nabla_{\boldsymbol{x}_t} \log p(\boldsymbol{x}_t|\boldsymbol{x}_s)|\boldsymbol{x}_t) = \mathrm{Tr}\,\mathrm{Cov}(\nabla_{\boldsymbol{x}_t} \log p(\boldsymbol{x}_t|\boldsymbol{x}_0)|\boldsymbol{x}_t) + d\,\kappa(s, t)$, *where $d$ is the data dimension. The map $s \mapsto \kappa(s, t)$ is strictly increasing on $[0, t)$, with $\kappa(0, t) = 0$.*

The proof is in Appendix C. The theorem establishes that noisy-target DSM is unbiased and shares the same population minimizer as clean-target DSM. However, its target has strictly larger conditional covariance (by $d\,\kappa(s, t)$), which increases the objective variance and slows optimization (Wang

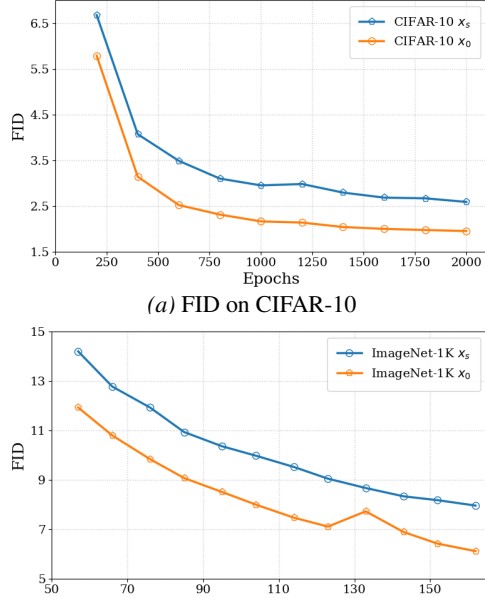

*(a)* FID on CIFAR-10

*(b)* FID on ImageNet-64

*Figure 3.* **Noisy target increases variance and hinders pretraining.** We compare the EDM baseline ($x_0$, clean-data objective; Equation (11)) with a noisy-target ($x_s$; Equation (12)). Under the same constraints, the noisy objective yields consistently worse FID on CIFAR-10 and ImageNet-64.

et al., 2013). We empirically corroborate this prediction with pretraining experiments on CIFAR-10 and ImageNet-64 using the EDM (Karras et al., 2022)[3] codebase, and provide a direct target-variance measurement in Table 6. The baseline uses the clean-data DSM objective Equation (11), where $\boldsymbol{D}_{\boldsymbol{\theta}}$ denotes the data prediction network:

$$
\mathbb{E}_{\boldsymbol{x}_0, \boldsymbol{x}_t} \left[ w(t) \left\| \boldsymbol{D}_{\boldsymbol{\theta}}(\boldsymbol{x}_t, t) - \boldsymbol{x}_0 \right\|^2 \right]
\tag{11}
$$

Under the $\boldsymbol{x}_0$-prediction framework (consistent with EDM), the noisy-target DSM objective is formulated as Equation (12), which mirrors the DDPO objective. The detailed derivation is provided in the Appendix C.5:

$$
\mathbb{E}_{\boldsymbol{x}_s, \boldsymbol{x}_t} \left[ w(t) \left\| \boldsymbol{D}_{\boldsymbol{\theta}}(\boldsymbol{x}_t, t) - \left( \boldsymbol{x}_t - \frac{t^2}{t^2 - s^2}(\boldsymbol{x}_t - \boldsymbol{x}_s) \right) \right\|^2 \right]
\tag{12}
$$

Under identical architectures, noise schedules, and optimization settings, the noisy-DSM objective learns in the correct direction but reaches a given quality level substantially later (see Figure 3), consistent with the variance analysis.

## 3. Advantage Weighted Matching

Motivated by our finding that DDPO implicitly optimizes DSM with *noisy* data, which increases variance, we introduce Advantage Weighted Matching (AWM), replacing the

---

[3]EDM's noise scheduler assumes that $\boldsymbol{x}_t = \boldsymbol{x}_0 + t\epsilon$, $t \in [0.002, 80]$ during inference and $t \in [0, +\infty]$ during training.

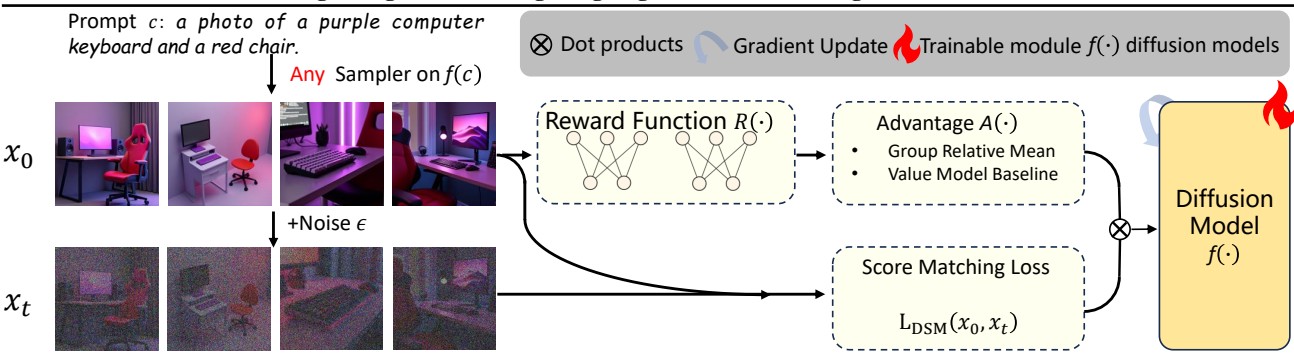

Figure 4. **AWM pipeline.** For each prompt $c$, we sample a group of sequences $\{x_0\}$ from a diffusion model, evaluate rewards, advantages, and train the sequence policy via advantage-weighted score matching. During inference, any ODE/SDE sampler can be used.

DDPO objective with DSM evaluated on *clean* data. Conceptually, this converts a step-wise reverse-transition policy into a sequence-level conditional policy over $x_0$. The resulting difference in problem setup is:

**Problem Setup Comparison.**

|  | DDPO | AWM |
|---|---|---|
| State | $(c, t, x_t)$ | $c$ |
| Action | $x_{t-1}$ | $x_0$ |
| Policy | $p_{\boldsymbol{\theta}}(x_{t-1} \mid x_t, c)$ | $p_{\boldsymbol{\theta}}(x_0 \mid c)$ |

As shown in Figure 4, for each prompt $c$, we sample a group $\{x_i\}_{i=1}^{G} \sim \pi_{\boldsymbol{\theta}_{\mathrm{old}}}(\cdot|c)$, compute rewards $r_i = r(x_i, c)$ and advantages $A_i$ (e.g., through group-relative mean $A_i = \frac{r_i - \mathrm{mean}(r_i)}{\mathrm{std}(r_i)}$, where $\mathrm{std}(r_i)$ can be computed either globally across the batch or locally within each group; we adopt the former to avoid introducing additional bias in advantage normalization, as discussed in recent analyses (Liu et al., 2025c)). The final objective for the Advantage Weighted Matching under the GRPO framework is as follows, where $\pi_{\boldsymbol{\theta}_{\mathrm{old}}}$ is the policy used to collect the sampling data, used for computing importance sampling ratios in GRPO/PPO, and $\pi_{\boldsymbol{\theta}_{\mathrm{ref}}}$ is the frozen reference model (usually the pretrained base model), used to calculate the KL divergence penalty to prevent reward hacking. In practice, the reference policy can also be periodically updated (Liu et al., 2025b) or implemented as an exponential moving average (EMA) of the training policy (Grill et al., 2020).

$$\mathcal{J}_{GRPO}(\boldsymbol{\theta}) = \mathbb{E}_{c, x_i} \frac{1}{G} \sum_{i=1}^{G} \left( \frac{\pi_{\boldsymbol{\theta}}(x_i|c)}{\pi_{\boldsymbol{\theta}_{\mathrm{old}}}(x_i|c)} \cdot A_i \right) \\ - \beta \mathbb{D}_{KL}(\pi_{\boldsymbol{\theta}} \| \pi_{\boldsymbol{\theta}_{\mathrm{ref}}}). \tag{13}$$

For simplicity, we omit the clip operation. We view the policy log-likelihood $\log \pi_{\boldsymbol{\theta}}(x_0|c)$ through its ELBO surrogate given by score/flow matching (Song et al., 2021; Kingma et al., 2021; Kingma & Gao, 2023):

$$-\mathbb{E}\left[w(t) \left\| v_{\boldsymbol{\theta}}(x_t, t, c) - (\epsilon - x_0) \right\|^2 \right], \tag{14}$$

where $w(t)$ is the ELBO time-weight $\frac{1-t}{t}$. In practice, we find that simple weight $w(t) = 1$ or $w(t) = t$ works better.

**Algorithm 1** Pseudo code of Advantage Weighted Matching

```
1   for i in range(T): # T: total steps
2       x = sampler(M, c) # M: model, c: prompt
3       r = reward_fn(x)  # r: reward, x: samples
4       A = cal_adv(r, c) # A: advantage
5
6       e = randn_like(x) # e: noise
7       t = get_timesteps(x)  # t: timesteps
8       x_t = fwd_diffusion(x, e, t) # x_t: noisy samples
9       v = M(x_t, t, c)  # v: velocity prediction
10      v_ref = M_ref(x_t, t, c) # v_ref: reference velocity
11
12      log_p = -((v - (e - x))**2).mean() # Flow Matching
            ↪ Loss
13      rho = torch.exp(log_p - log_p.detach())
14      L_policy = -A * rho # policy loss
15      L_kl = w(t) * ((v - v_ref)**2).mean() # KL loss
16      L = L_policy + beta * L_kl # total loss
```

This discrepancy reflects the different nature of the visual quality and likelihood for diffusion models, and has been widely observed in prior work (Ho et al., 2020; Nichol & Dhariwal, 2021; Song et al., 2021; Vahdat et al., 2021; Watson et al., 2021; Kingma et al., 2021). Thus, the likelihood ratio $\frac{\pi_{\boldsymbol{\theta}}(x|c)}{\pi_{\boldsymbol{\theta}_{\mathrm{old}}}(x|c)}$ can be estimated through:

$$\exp(-(\ell_{\boldsymbol{\theta}} - \ell_{\boldsymbol{\theta}_{\mathrm{old}}})), \tag{15}$$

where $\ell_{\boldsymbol{\theta}} = \mathbb{E}\left[w(t) \|v_{\boldsymbol{\theta}}(x_t, t, c) - (\epsilon - x_0)\|^2 \right]$. The KL term can be estimated (Song et al., 2021) through:

$$\mathbb{E}\left[w(t) \|v_{\boldsymbol{\theta}}(x_t, t, c) - v_{\boldsymbol{\theta}_{\mathrm{ref}}}(x_t, t, c)\|^2 \right]. \tag{16}$$

To make the mechanics explicit, consider an *on-policy* update on a sample $x_i$ drawn from $\pi_{\theta}(\cdot \mid c)$. Let $\hat{\pi}_{\theta}$ denote the ELBO-based surrogate of the likelihood,

$$\nabla_{\boldsymbol{\theta}} \frac{\hat{\pi}_{\boldsymbol{\theta}}(x_i|c)}{\mathrm{stopgrad}\left(\hat{\pi}_{\boldsymbol{\theta}}(x_i|c)\right)} \cdot A_i = \nabla_{\boldsymbol{\theta}} \log \hat{\pi}_{\boldsymbol{\theta}}(x_i|c) \cdot A_i$$
$$= -\nabla_{\boldsymbol{\theta}} \mathbb{E}\left[w(t) \|v_{\boldsymbol{\theta}}(x_t, t, c) - (\epsilon - x_0)\|^2 \right] \cdot A_i. \tag{17}$$

When $A_i > 0$ (good sample), the gradient decreases the flow matching loss at $(x_i, c)$, pulling $v_{\theta}$ toward the target $(\epsilon - x_i)$; when $A_i < 0$ (bad sample), the gradient will push the velocity away from the undesirable target.

*Table 1.* **Performance comparison on GenEval.** Speed-up is relative to Flow-GRPO baseline.

| Model | GenEval | | | | | | | |
|---|---|---|---|---|---|---|---|---|
| | Single Obj. | Two Obj. | Counting | Color | Position | Attr | Overall | Speed-up |
| DALLE-3 | 0.96 | 0.87 | 0.47 | 0.83 | 0.43 | 0.45 | 0.67 | – |
| GPT-4o | 0.99 | 0.92 | 0.85 | 0.92 | 0.75 | 0.61 | 0.84 | – |
| SD-XL | 0.98 | 0.74 | 0.39 | 0.85 | 0.15 | 0.23 | 0.55 | – |
| FLUX.1 Dev | 0.98 | 0.81 | 0.74 | 0.79 | 0.22 | 0.45 | 0.66 | – |
| SD3.5L | 0.98 | 0.89 | 0.73 | 0.83 | 0.34 | 0.47 | 0.71 | – |
| SD3.5M | 0.98 | 0.78 | 0.50 | 0.81 | 0.24 | 0.52 | 0.63 | – |
| Flow-GRPO | **1.00** | 0.99 | 0.95 | 0.92 | 0.99 | 0.86 | 0.95 | 1× |
| TempFlow-GRPO | **1.00** | **1.00** | 0.96 | 0.95 | **0.99** | 0.91 | 0.97 | 2.8× |
| AWM (Ours) | **1.00** | **1.00** | **0.99** | **0.97** | **0.99** | **0.94** | **0.985** | **16.54×** |

*(a)* SD3.5M OCR  *(b)* SD3.5M PickScore  *(c)* FLUX OCR  *(d)* FLUX PickScore

*Figure 5.* **OCR and PickScore training efficiency.** Metric vs. GPU hours for SD3.5M and FLUX. AWM (ours) exceeds Flow-GRPO with far less compute.

We provide a clear pseudo-code Algorithm 1 and pipeline Figure 4 for our AWM algorithm. For each prompt $c$, we (i) sample a group of samples $\{x_0\}_{i=1}^G$, (ii) evaluate rewards and form group-relative advantages $A_i$, (iii) get knoisy data $x_t = (1-t)x_0 + t\epsilon$ to evaluate score matching losses, and (iv) optimize an *advantage-weighted* FM objective with a velocity-space KL regularization term.

We summarize several key differences between AWM and DDPO below:

**Variance of Policy Gradient.** Subtracting a baseline in policy gradient will keep the gradient unbiased and reduce the variance as an additive control variate (Greensmith et al., 2004). For diffusion models, there is another level of variance that comes from the estimation of the likelihood as the exact likelihood is computationally intractable. AWM reduces the variance compared to DDPO.

**Decouple of Sampling and Training.** The forward process formulation decouples the training and sampling procedure of AWM. This has several advantages: First, the samplers are not restricted to DDPM or Euler-Maruyama; we can use more advanced ODE samplers (Lu et al., 2022) or SDE samplers (Xue et al., 2023) for a better trade-off between quality and speed. Second, the training timesteps and sampling timesteps can be decoupled, *e.g.*, we can use 20 steps for sampling and 4 steps for training. Third, this also allows future explorations, such as using the step-distilled model to accelerate sampling.

**Alignment with Pretraining.** AWM keeps the conceptual symmetry between pretraining and RL for diffusion as in LLM: both optimize the same DSM/FM loss, differing only by *reward-derived weights*. Also, this avoids the use of CFG (Ho & Salimans, 2022) in training as pretraining.

## 4. Experiments

We use two representative open-source models: SD3.5M (Esser et al., 2024) and FLUX (Labs, 2024) on the above three reward tasks, i.e., composition image generation on GenEval (Ghosh et al., 2023), visual-text rendering on OCR (Chen et al., 2023a), and human preference alignment on PickScore (Kirstain et al., 2023). We use a group size $G = 24$, and LoRA (Hu et al., 2022) with $\alpha = 64$ and $r = 32$ for SD3.5M and LoRA with $\alpha = 128$ and $r = 64$ for FLUX. The learning rate is set to a constant $3e - 4$. We use $w(t) = t$ in Equation (15), together with SA-Solver (Xue et al., 2023) with noise level $\eta = 0.4$, 14 timesteps, and a guidance scale of cfg = 1.0.

We use 6 Monte Carlo samples to estimate the ELBO. We set the reference model to an Exponential Moving Average of the running model, with the EMA rate set as $\eta = \min\{0.001t, 0.3\}$, where $t$ is the iter number. The KL ratio $\beta$ is set to 1 for GenEval, OCR, and PickScore. We maintain a purely on-policy sampling-training regime, where each generated batch is used for a single gradient update without off-policy reuse. We provide additional early experiment settings and results in Appendix F.

*Table 2.* **Performance comparison on OCR/PickScore for SD3.5M and FLUX.** Hours is the total GPU hours; Speed-up is relative to Flow-GRPO baseline. [†] indicates longer training time.

| Method | SD3.5M | | | | FLUX | | | |
|---|---|---|---|---|---|---|---|---|
| | OCR | | PickScore | | OCR | | PickScore | |
| | Acc | Hours | Score | Hours | Acc | Hours | Score | GPU hours |
| Base model | 0.59 | – | 21.72 | – | 0.59 | – | 22.20 | – |
| FlowGRPO | 0.89 | 415.9 (1×) | 23.01 | 956.1 (1×) | 0.95 | 343.6 (1×) | 23.08 | 339.2 (1×) |
| AWM (Ours) | 0.92 | 21.76 (19.13×) | 23.11 | 27.56 (34.67×) | 0.95 | 40.3 (8.53×) | 23.08 | 49.8 (6.82×) |
| AWM (Ours)[†] | 0.96(+7.87%) | 58.02 | 24.03 (+4.43%) | 992.0 | 0.99 (+4.21%) | 147.0 | 23.18 (+0.43%) | 78.0 |

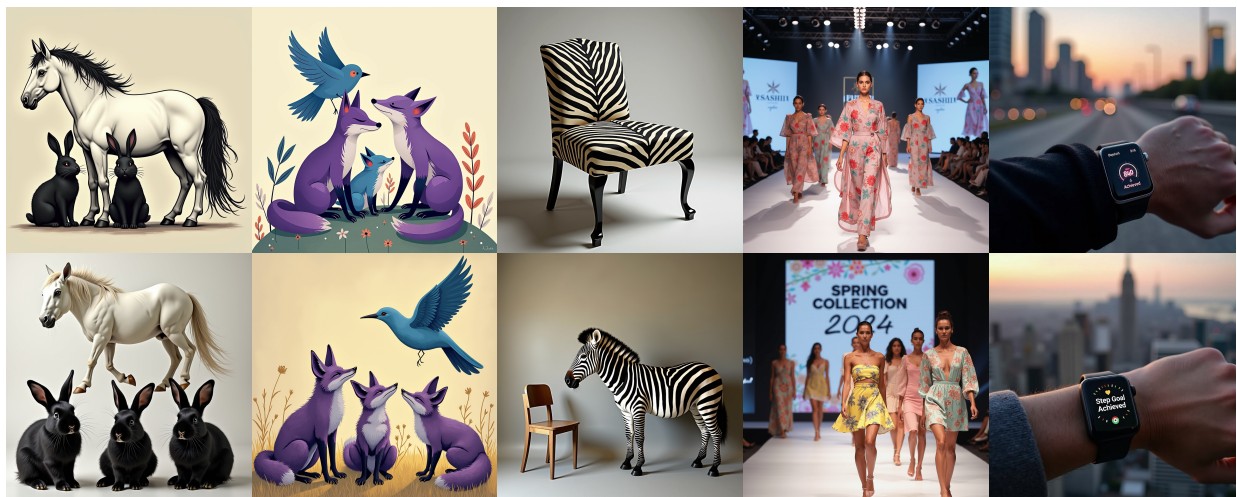

*Figure 6.* **Visual comparison** before (first row) and after 100 steps of AWM training (second row). Prompts from the GenEval and OCR benchmarks (listed in [4]) are used for generation.

## 4.1. Main Results

Table 1 reports the comparative performance of various models on the GenEval benchmark for SD3.5M. Among popular models, e.g., SD3.5L, DALLE-3, GPT-4o, and FLUX.1 Dev, overall scores range between 0.55 and 0.84, with GPT-4o achieving the highest performance (0.84). After RL post-training, Flow-GRPO reaches an overall GenEval score of 0.95, serving as the baseline. AWM reaches the same performance level 16.54× faster in terms of GPU hours, and continues to improve, ultimately achieving a higher final GenEval score of 0.985. Notably, AWM also attains consistently strong results across all sub-tasks, including Color Attribution (0.94), Counting (0.99), confirming that it preserves quality while substantially reducing training cost.

Besides, we further validated AWM on other metrics such as OCR and PickScore, and across different backbones, including SD3.5M and FLUX. As shown in Table 2, AWM surpasses Flow-GRPO in OCR and PickScore while requiring far fewer GPU hours. For example, SD3.5M with AWM achieves OCR (0.92) with a 19.13× speed-up, and OCR (0.95) with an 8.53× reduction on FLUX.

We illustrate training efficiency in Figure 5. AWM achieves superior performance on both OCR and PickScore while using significantly fewer GPU hours—for example, up to

34.67× faster on SD3.5M PickScore and 6.82× faster on FLUX PickScore. These results demonstrate consistent efficiency gains over Flow-GRPO across models and evaluation metrics. Specifically, AWM reaches an OCR score of 0.96 on SD3.5M using only 58 GPU hours, whereas Flow-GRPO requires over 400 GPU hours to reach an OCR score of 0.90.

We visualize the FLUX baseline and FLUX after 100 AWM gradient steps on three composition prompts and two text-rendering (OCR) prompts. The composition prompts specify numerosity, color, and position constraints. As shown in the second column of Figure 6, AWM adheres closely to the instructions—e.g., "three," "purple," "fox," "one," "blue"—and likewise improves text rendering.

## 4.2. Ablation Studies

Unless otherwise stated, all ablations are conducted on SD3.5M with identical architectures, batch sizes, and reward pipelines; curves report GenEval score versus training steps. Results are summarized in Figure 7 and Figure 8.

---

[4]Prompts in Figure 6 are: (1) three black rabbits below one white horse; (2) three purple foxes below one blue bird; (3) a photo of a chair left of a zebra; (4) A high-fashion runway with a sleek, modern backdrop displaying **"Spring Collection 2024"**. Models walk confidently on the catwalk, showcasing vibrant, floral prints and pastel tones, under soft, ambient lighting that enhances the fresh, spring vibe; (5) A close-up of a sleek smartwatch on a wrist, the screen displaying **"Step Goal Achieved"** with a celebratory animation, set against a blurred cityscape at dusk, capturing the moment of accomplishment.

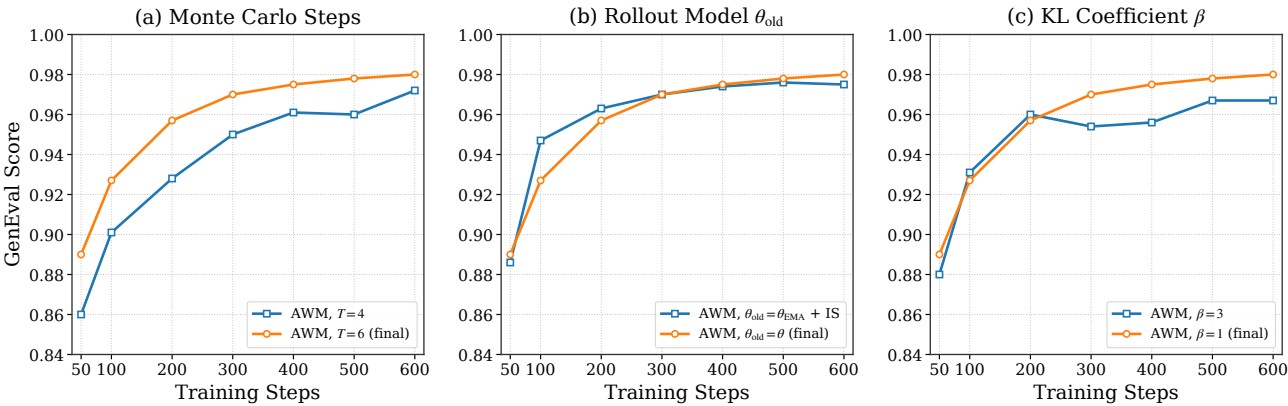

*Figure 7.* **Ablations on SD3.5M and GenEval.** (a) Number of Monte Carlo steps $T$; (b) rollout model $\theta_{\mathrm{old}}$, where $\theta_{\mathrm{EMA}}$ + IS denotes sampling from an EMA copy of the online model and applying the importance-sampling correction in Equation (15); (c) KL regularization coefficient $\beta$. Curves show GenEval score vs. training steps.

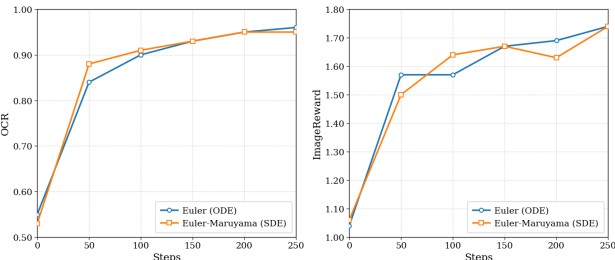

   *(a)* OCR: ODE vs. SDE     *(b)* ImageReward: ODE vs. SDE

*Figure 8.* **ODE vs. SDE sampling under different reward models.** Comparison of deterministic (ODE) and stochastic (SDE) samplers when optimizing OCR and ImageReward objectives. Curves show reward performance versus GPU hours.

**Number of Monte Carlo steps $T$.** We compare using $T = 4$ and $T = 6$ Monte Carlo timesteps to estimate the AWM objective. As shown in Figure 7a, both settings learn stably, while $T = 6$ gives a stronger final GenEval score. We therefore use $T = 6$ as the default setting.

**Rollout model $\theta_{\mathbf{old}}$.** We compare the standard on-policy rollout, $\theta_{\mathrm{old}} = \mathrm{sg}(\theta)$, with an EMA rollout policy denoted $\theta_{\mathrm{EMA}}$+ Importance Sampling (IS). In the EMA variant, samples are generated by an exponential-moving-average copy of the online model, updated with rate $\eta_{\mathrm{EMA}}(n) = \min\{0.001n, 0.3\}$ at training iteration $n$; because these samples are off-policy relative to the current model, we optimize them with the importance-sampling ratio in Equation (15). Figure 7b shows that the EMA+IS variant is competitive, but the on-policy setting achieves a slightly better final score with a simpler training loop. We thus adopt on-policy rollouts in experiments.

**KL regularization coefficient $\beta$.** We compare $\beta = 1$ and $\beta = 3$ in the velocity-space KL proxy. As shown in Figure 7c, stronger regularization remains stable but slows learning and underperforms the lighter setting. We therefore set $\beta = 1$ by default, so that KL regularization stabilizes the update without dominating the reward signal.

**Compatibility across samplers.** AWM operates purely on the final samples $x_0$ and does not rely on any sampler-specific trajectories. To further validate this property, we additionally compare deterministic ODE sampling and stochastic SDE sampling under the same training setup. As shown in Figure 8, both samplers yield highly similar reward performance across different numbers of steps, for both OCR and ImageReward objectives.

## 5. Conclusion

In this paper, we showed that DDPO implicitly performs denoising score/flow matching with noisy targets, which increases variance and slows optimization. Building on this, we introduced *Advantage-Weighted Matching* (AWM), which keeps the pretraining score/flow-matching objective and applies advantage-based weights—thereby unifying pretraining and RL post-training under a single, policy-gradient–consistent objective. It achieves up to $34\times$ faster convergence than Flow-GRPO on SD3.5M and FLUX without degrading generation quality.

## Impact Statement

This work advances the theoretical understanding of reinforcement learning–based optimization for diffusion models by clarifying the relationship between DDPO and denoising score matching and analyzing the variance effects induced by noisy targets. The proposed analysis and the Advantage Weighted Matching framework may enable more stable and efficient training of diffusion models under diverse reward functions and sampling procedures. Improved training efficiency can help reduce computational cost and energy consumption in large-scale generative modeling. As with other advances in generative models, the techniques studied here could be misused to generate misleading or harmful content, a risk that is not unique to this work. Addressing such concerns requires responsible deployment and safeguards beyond the scope of this paper.

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

*Table 3.* Notations utilized in this paper.

| Notation | Description |
| --- | --- |
| $\boldsymbol{x}_0$ | Clean data sample from the data distribution |
| $\boldsymbol{x}_t$ | Noisy data at time $t$ |
| $\boldsymbol{c}$ | Conditioning context (e.g., text prompt) |
| $\boldsymbol{\epsilon}$ | Standard Gaussian noise, $\boldsymbol{\epsilon} \sim \mathcal{N}(\mathbf{0}, \boldsymbol{I})$ |
| $t$ | Time step, $t \in [0, 1]$ |
| $\boldsymbol{v}_{\boldsymbol{\theta}}(\boldsymbol{x}_t, t)$ | Velocity prediction network |
| $\boldsymbol{\epsilon}_{\boldsymbol{\theta}}(\boldsymbol{x}_t, t)$ | Noise prediction network |
| $\boldsymbol{s}_{\boldsymbol{\theta}}(\boldsymbol{x}_t, t)$ | Score prediction network |
| $p(\boldsymbol{x})$ | Likelihood of the data |
| $\Delta t$ | Discretization step size for SDE/ODE solvers |
| $\pi_{\boldsymbol{\theta}}$ | Policy model parameterized by $\boldsymbol{\theta}$ |
| $\pi_{\boldsymbol{\theta}_{\text{old}}}$ | Old policy used for sampling |
| $\pi_{\boldsymbol{\theta}_{\text{ref}}}$ | Reference policy for KL regularization (e.g. base model) |
| $r(\boldsymbol{x}, \boldsymbol{c})$ | Reward function evaluating the quality of sample $\boldsymbol{x}$ given $\boldsymbol{c}$ |
| $A$ | Advantage value derived from rewards |
| $G$ | Group size of samples generated per prompt |
| $\mathcal{J}_{GRPO}$ | Objective function for GRPO |
| $\mathbb{D}_{KL}$ | Kullback-Leibler divergence |
| $\beta$ | Coefficient for the KL regularization term |
| $w(t)$ | Time-dependent weighting function in the loss objective |

## A. Notation

In this section, we compile a comprehensive list of mathematical notations used throughout the paper to facilitate reference. Given that our work, Advantage Weighted Matching (AWM), bridges the domains of generative diffusion models (specifically Flow Matching) and Reinforcement Learning (RL), we explicitly define symbols for both the generative processes and the policy optimization objectives. A detailed summary of these notations and their corresponding descriptions is provided in Table 3.

## B. LLM Usage

In the preparation of this manuscript, we employed large language models to provide language-related assistance. Specifically, the LLM was used to (i) polish grammar, style, and readability of the text; (ii) offer suggestions for clearer phrasing and more concise expression.

# C. Proofs

## C.1. Proof of Lemma 2.2

*Proof.* The proof proceeds by taking the gradient of the Denoising Score Matching loss with respect to the model parameters $\boldsymbol{\theta}$ and showing that it is equivalent to the gradient of the standard Score Matching loss.

$$
\begin{aligned}
&\nabla_{\boldsymbol{\theta}} \mathbb{E}_{\boldsymbol{x}_s} \mathbb{E}_{\boldsymbol{x}_t|\boldsymbol{x}_s} \left[ \|\boldsymbol{s}_{\boldsymbol{\theta}}(\boldsymbol{x}_t, t) - \nabla_{\boldsymbol{x}_t} \log p(\boldsymbol{x}_t|\boldsymbol{x}_s)\|^2 \right] \\
=&\nabla_{\boldsymbol{\theta}} \mathbb{E}_{\boldsymbol{x}_s} \mathbb{E}_{\boldsymbol{x}_t|\boldsymbol{x}_s} \left[ \|\boldsymbol{s}_{\boldsymbol{\theta}}(\boldsymbol{x}_t, t)\|^2 \right] - 2\nabla_{\boldsymbol{\theta}} \mathbb{E}_{\boldsymbol{x}_s} \mathbb{E}_{\boldsymbol{x}_t|\boldsymbol{x}_s} \left[ \langle \boldsymbol{s}_{\boldsymbol{\theta}}(\boldsymbol{x}_t, t), \nabla_{\boldsymbol{x}_t} \log p(\boldsymbol{x}_t|\boldsymbol{x}_s) \rangle \right] \\
=&\nabla_{\boldsymbol{\theta}} \mathbb{E}_{\boldsymbol{x}_t} \left[ \|\boldsymbol{s}_{\boldsymbol{\theta}}(\boldsymbol{x}_t, t)\|^2 \right] - 2\nabla_{\boldsymbol{\theta}} \int_{\boldsymbol{x}_t} \int_{\boldsymbol{x}_s} \langle \boldsymbol{s}_{\boldsymbol{\theta}}(\boldsymbol{x}_t, t), \nabla_{\boldsymbol{x}_t} \log p(\boldsymbol{x}_t|\boldsymbol{x}_s) \rangle p(\boldsymbol{x}_t|\boldsymbol{x}_s) p(\boldsymbol{x}_s) \mathrm{d}\boldsymbol{x}_s \mathrm{d}\boldsymbol{x}_t \\
=&\nabla_{\boldsymbol{\theta}} \mathbb{E}_{\boldsymbol{x}_t} \left[ \|\boldsymbol{s}_{\boldsymbol{\theta}}(\boldsymbol{x}_t, t)\|^2 \right] - 2\nabla_{\boldsymbol{\theta}} \int_{\boldsymbol{x}_t} \int_{\boldsymbol{x}_s} \langle \boldsymbol{s}_{\boldsymbol{\theta}}(\boldsymbol{x}_t, t), \nabla_{\boldsymbol{x}_t} p(\boldsymbol{x}_t|\boldsymbol{x}_s) \rangle p(\boldsymbol{x}_s) \mathrm{d}\boldsymbol{x}_s \mathrm{d}\boldsymbol{x}_t \\
=&\nabla_{\boldsymbol{\theta}} \mathbb{E}_{\boldsymbol{x}_t} \left[ \|\boldsymbol{s}_{\boldsymbol{\theta}}(\boldsymbol{x}_t, t)\|^2 \right] - 2\nabla_{\boldsymbol{\theta}} \int_{\boldsymbol{x}_t} \langle \boldsymbol{s}_{\boldsymbol{\theta}}(\boldsymbol{x}_t, t), \int_{\boldsymbol{x}_s} \nabla_{\boldsymbol{x}_t} p(\boldsymbol{x}_t|\boldsymbol{x}_s) p(\boldsymbol{x}_s) \mathrm{d}\boldsymbol{x}_s \rangle \mathrm{d}\boldsymbol{x}_t \\
=&\nabla_{\boldsymbol{\theta}} \mathbb{E}_{\boldsymbol{x}_t} \left[ \|\boldsymbol{s}_{\boldsymbol{\theta}}(\boldsymbol{x}_t, t)\|^2 \right] - 2\nabla_{\boldsymbol{\theta}} \int_{\boldsymbol{x}_t} \langle \boldsymbol{s}_{\boldsymbol{\theta}}(\boldsymbol{x}_t, t), \nabla_{\boldsymbol{x}_t} \int_{\boldsymbol{x}_s} p(\boldsymbol{x}_t|\boldsymbol{x}_s) p(\boldsymbol{x}_s) \mathrm{d}\boldsymbol{x}_s \rangle \mathrm{d}\boldsymbol{x}_t \\
=&\nabla_{\boldsymbol{\theta}} \mathbb{E}_{\boldsymbol{x}_t} \left[ \|\boldsymbol{s}_{\boldsymbol{\theta}}(\boldsymbol{x}_t, t)\|^2 \right] - 2\nabla_{\boldsymbol{\theta}} \int_{\boldsymbol{x}_t} \langle \boldsymbol{s}_{\boldsymbol{\theta}}(\boldsymbol{x}_t, t), \nabla_{\boldsymbol{x}_t} p(\boldsymbol{x}_t) \rangle \mathrm{d}\boldsymbol{x}_t \\
=&\nabla_{\boldsymbol{\theta}} \mathbb{E}_{\boldsymbol{x}_t} \left[ \|\boldsymbol{s}_{\boldsymbol{\theta}}(\boldsymbol{x}_t, t)\|^2 \right] - 2\nabla_{\boldsymbol{\theta}} \int_{\boldsymbol{x}_t} \langle \boldsymbol{s}_{\boldsymbol{\theta}}(\boldsymbol{x}_t, t), \nabla_{\boldsymbol{x}_t} \log p(\boldsymbol{x}_t) \rangle p(\boldsymbol{x}_t) \mathrm{d}\boldsymbol{x}_t \\
=&\nabla_{\boldsymbol{\theta}} \mathbb{E}_{\boldsymbol{x}_t} \left[ \|\boldsymbol{s}_{\boldsymbol{\theta}}(\boldsymbol{x}_t, t)\|^2 \right] - 2\nabla_{\boldsymbol{\theta}} \mathbb{E}_{\boldsymbol{x}_t} \left[ \langle \boldsymbol{s}_{\boldsymbol{\theta}}(\boldsymbol{x}_t, t), \nabla_{\boldsymbol{x}_t} \log p(\boldsymbol{x}_t) \rangle \right] \\
=&\nabla_{\boldsymbol{\theta}} \mathbb{E}_{\boldsymbol{x}_t} \left[ \|\boldsymbol{s}_{\boldsymbol{\theta}}(\boldsymbol{x}_t, t) - \nabla_{\boldsymbol{x}_t} \log p(\boldsymbol{x}_t)\|^2 \right]
\end{aligned}
\tag{18}
$$

$\square$

## C.2. Proof of Theorem 2.1

Our analysis builds upon a foundational theorem regarding the time reversal of diffusion processes from Haussmann and Pardoux (Haussmann & Pardoux, 1986).

**Lemma C.1** (Time Reversal of Diffusions (Haussmann & Pardoux, 1986)). *Let the forward process $\{\boldsymbol{x}_t\}_{t \in [0,1]}$ be defined by the stochastic differential equation (SDE):*

$$
\mathrm{d}\boldsymbol{x}_t = -\frac{1}{1-t} \boldsymbol{x}_t \mathrm{d}t + \sqrt{\frac{2t}{1-t}} \mathrm{d}\boldsymbol{w}_t,
\tag{19}
$$

*Then, the corresponding reverse process, where time flows from 1 to 0, satisfies the SDE:*

$$
\mathrm{d}\boldsymbol{x}_t = \left[ \boldsymbol{v}_{\boldsymbol{\theta}}(\boldsymbol{x}_t, t) + \frac{1}{1-t} \left( \boldsymbol{x}_t + (1-t)\boldsymbol{v}_{\boldsymbol{\theta}}(\boldsymbol{x}_t, t) \right) \right] \mathrm{d}t + \sqrt{\frac{2t}{1-t}} \mathrm{d}\boldsymbol{w}_t.
\tag{20}
$$

A direct consequence of this theorem is that the marginal distribution of $\boldsymbol{x}_t$ is identical for the forward and reverse processes. Furthermore, a stronger corollary holds: the joint distribution of $(\boldsymbol{x}_s, \boldsymbol{x}_t)$ is also identical for both processes. This allows us to connect the forward-process-based score matching objective to the reverse-process-based DDPO objective.

*Proof.* Our goal is to demonstrate that minimizing the Denoising Score Matching loss is equivalent to maximizing the DDPO log-likelihood objective. We begin with the Denoising Score Matching loss at $\boldsymbol{x}_t$ conditioned on noisy data $\boldsymbol{x}_{t-\Delta t}$ from the forward process:

$$
\mathbb{E}_{\boldsymbol{x}_{t-\Delta t}, \boldsymbol{x}_t} \left[ \|\boldsymbol{s}_{\boldsymbol{\theta}}(\boldsymbol{x}_t, t) - \nabla \log p(\boldsymbol{x}_t|\boldsymbol{x}_{t-\Delta t})\|^2 \right].
\tag{21}
$$

Note that $\boldsymbol{x}_t | \boldsymbol{x}_{t-\Delta t} \sim \mathcal{N}\left(\frac{1-t}{1-(t-\Delta t)}\boldsymbol{x}_{t-\Delta t}, \left(t^2 - \frac{(1-t)^2}{(1-(t-\Delta t))^2}(t - \Delta t)^2\right)\boldsymbol{I}\right)$, thus we have that the above denoising score matching loss is:

$$\mathbb{E}_{\boldsymbol{x}_{t-\Delta t}, \boldsymbol{x}_t}\left[\left\|\boldsymbol{s}_{\boldsymbol{\theta}}(\boldsymbol{x}_t, t) + \frac{\boldsymbol{x}_t - \frac{1-t}{1-(t-\Delta t)}\boldsymbol{x}_{t-\Delta t}}{t^2 - \frac{(1-t)^2}{(1-(t-\Delta t))^2}(t - \Delta t)^2}\right\|^2\right]. \tag{22}$$

Also, note that $\boldsymbol{s}_{\boldsymbol{\theta}}(\boldsymbol{x}_t, t) = -\frac{(1-t)\boldsymbol{v}_{\boldsymbol{\theta}}(\boldsymbol{x}_t, t) + \boldsymbol{x}_t}{t}$, we have

$$\mathbb{E}_{\boldsymbol{x}_{t-\Delta t}, \boldsymbol{x}_t}\left[\left\|-\frac{(1-t)\boldsymbol{v}_{\boldsymbol{\theta}}(\boldsymbol{x}_t, t) + \boldsymbol{x}_t}{t} + \frac{\boldsymbol{x}_t - \frac{1-t}{1-(t-\Delta t)}\boldsymbol{x}_{t-\Delta t}}{t^2 - \frac{(1-t)^2}{(1-(t-\Delta t))^2}(t - \Delta t)^2}\right\|^2\right]. \tag{23}$$

Simplifying the above equation, and let the coefficient of $\boldsymbol{x}_{t-\Delta t}$ to be 1, we have:

$$\mathbb{E}_{\boldsymbol{x}_{t-\Delta t}, \boldsymbol{x}_t}\left[\left\|\boldsymbol{x}_{t-\Delta t} - \left((1 - (t - \Delta t)) + \frac{(1-t)(t - \Delta t)^2}{t(1 - (t - \Delta t))}\right)\boldsymbol{x}_t\right.\right.$$
$$\left.\left. + \frac{1 - (t - \Delta t)}{t}\left(t^2 - \frac{(1-t)^2}{(1 - (t - \Delta t))^2}(t - \Delta t)^2\right)\boldsymbol{v}_{\boldsymbol{\theta}}(\boldsymbol{x}_t, t)\right\|^2\right]. \tag{24}$$

Now, let's consider the DDPO objective, which is the log-likelihood of one step of the reverse process. We consider the Euler-Maruyama discretization with Exponential Integrator of the diffusion SDE Equation (1):

$$d\boldsymbol{x}_t = \left[\boldsymbol{v}_{\boldsymbol{\theta}}(\boldsymbol{x}_t, t) + \frac{1}{1-t}(\boldsymbol{x}_t + (1-t)\boldsymbol{v}_{\boldsymbol{\theta}}(\boldsymbol{x}_t, t))\right]dt + \sqrt{\frac{2t}{1-t}}d\boldsymbol{w}_t. \tag{25}$$

According to the results in (Xue et al., 2023),

$$\boldsymbol{x}_{t-\Delta t} = \frac{(1-t)(t - \Delta t)^2}{(1 - (t - \Delta t))t^2}\boldsymbol{x}_t + \frac{1 - (t - \Delta t)}{t^2}(t^2 - \frac{(1-t)^2}{(1 - (t - \Delta t))^2}(t - \Delta t)^2)\boldsymbol{x}_{\boldsymbol{\theta}}(\boldsymbol{x}_t, t)$$
$$+ \sqrt{t^2 - \frac{(1-t)^2}{(1 - (t - \Delta t))^2}(t - \Delta t)^2}\frac{t - \Delta t}{t}\boldsymbol{\epsilon} \tag{26}$$

Substitute that $\boldsymbol{x}_{\boldsymbol{\theta}} = \boldsymbol{x}_t - t\boldsymbol{v}_{\boldsymbol{\theta}}$, we have that

$$\boldsymbol{x}_{t-\Delta t} = \left((1 - (t - \Delta t)) + \frac{(1-t)(t - \Delta t)^2}{t(1 - (t - \Delta t))}\right)\boldsymbol{x}_t \tag{27}$$
$$- \frac{1 - (t - \Delta t)}{t}(t^2 - \frac{(1-t)^2}{(1 - (t - \Delta t))^2}(t - \Delta t)^2)\boldsymbol{v}_{\boldsymbol{\theta}}(\boldsymbol{x}_t, t)$$
$$+ \sqrt{t^2 - \frac{(1-t)^2}{(1 - (t - \Delta t))^2}(t - \Delta t)^2}\frac{t - \Delta t}{t}\boldsymbol{\epsilon} \tag{28}$$

If we omit the discretization error induced by Euler-Maruyama discretization, according to Lemma C.1, the joint distribution of $(\boldsymbol{x}_{t-\Delta t}, \boldsymbol{x}_t)$ in the reverse process is identical to $(\boldsymbol{x}_{t-\Delta t}, \boldsymbol{x}_t)$ in the forward process. Then we have that maximizing $\log p_{\boldsymbol{\theta}}(\boldsymbol{x}_{t-\Delta t} | \boldsymbol{x}_t)$ is equivalent to minimizing:

$$\mathbb{E}_{\boldsymbol{x}_{t-\Delta t}, \boldsymbol{x}_t}\left[\left\|\boldsymbol{x}_{t-\Delta t} - \left((1 - (t - \Delta t)) + \frac{(1-t)(t - \Delta t)^2}{t(1 - (t - \Delta t))}\right)\boldsymbol{x}_t\right.\right.$$
$$\left.\left. + \frac{1 - (t - \Delta t)}{t}\left(t^2 - \frac{(1-t)^2}{(1 - (t - \Delta t))^2}(t - \Delta t)^2\right)\boldsymbol{v}_{\boldsymbol{\theta}}(\boldsymbol{x}_t, t)\right\|^2\right], \tag{29}$$

which is identical to Equation (24), thus identical to the noisy DSM loss Equation (21). $\square$

## C.3. Advantage-weighted Extension from DDPO Likelihood to Clean-target DSM

Theorem 2.1 identifies the DDPO per-step likelihood with noisy-target DSM, while Lemma 2.2 connects noisy-target DSM to clean-target DSM. This subsection extends this two-step connection to the local gradient of the advantage-weighted DDPO objective. Let $\bar{\boldsymbol{\theta}} := \boldsymbol{\theta}_{\texttt{old}} = \text{sg}(\boldsymbol{\theta})$ be the rollout policy, and let

$$A(\boldsymbol{x}_0, \boldsymbol{c}) = r(\boldsymbol{x}_0, \boldsymbol{c}) - b(\boldsymbol{c}) \tag{30}$$

be the terminal advantage. Importantly, we do not assume that $A(\boldsymbol{x}_0, \boldsymbol{c})$ is independent of $\boldsymbol{x}_t$.

Define the reverse-rollout law

$$Q_{\bar{\boldsymbol{\theta}}}^{\text{rev}}(\boldsymbol{c}, \boldsymbol{x}_{0:T}) := p(\boldsymbol{c})p(\boldsymbol{x}_T) \prod_{k=1}^{T} p_{\bar{\boldsymbol{\theta}}}(\boldsymbol{x}_{k-1}|\boldsymbol{x}_k, \boldsymbol{c}). \tag{31}$$

For a fixed adjacent pair $s = t - \Delta t$, isolate the $t$-th DDPO surrogate:

$$J_t^{\text{DDPO}}(\boldsymbol{\theta}; \bar{\boldsymbol{\theta}}) := \mathbb{E}_{Q_{\bar{\boldsymbol{\theta}}}^{\text{rev}}} \left[ A(\boldsymbol{x}_0, \boldsymbol{c}) \frac{p_{\boldsymbol{\theta}}(\boldsymbol{x}_s|\boldsymbol{x}_t, \boldsymbol{c})}{p_{\bar{\boldsymbol{\theta}}}(\boldsymbol{x}_s|\boldsymbol{x}_t, \boldsymbol{c})} \right]. \tag{32}$$

Taking the local gradient at the frozen policy gives

$$\nabla_{\boldsymbol{\theta}} J_t^{\text{DDPO}}(\boldsymbol{\theta}; \bar{\boldsymbol{\theta}})\big|_{\boldsymbol{\theta}=\bar{\boldsymbol{\theta}}} = \mathbb{E}_{Q_{\bar{\boldsymbol{\theta}}}^{\text{rev}}} \left[ A(\boldsymbol{x}_0, \boldsymbol{c}) \, \nabla_{\boldsymbol{\theta}} \log p_{\boldsymbol{\theta}}(\boldsymbol{x}_s|\boldsymbol{x}_t, \boldsymbol{c})\big|_{\boldsymbol{\theta}=\bar{\boldsymbol{\theta}}} \right], \tag{33}$$

since $\nabla_{\boldsymbol{\theta}}(p_{\boldsymbol{\theta}}/p_{\bar{\boldsymbol{\theta}}}) = (p_{\boldsymbol{\theta}}/p_{\bar{\boldsymbol{\theta}}})\nabla_{\boldsymbol{\theta}} \log p_{\boldsymbol{\theta}}$ and the ratio equals one at $\boldsymbol{\theta} = \bar{\boldsymbol{\theta}}$. The baseline term is unbiased because

$$\mathbb{E}_{\boldsymbol{c}, \boldsymbol{x}_t} \left[ b(\boldsymbol{c}) \int \nabla_{\boldsymbol{\theta}} p_{\boldsymbol{\theta}}(\boldsymbol{x}_s|\boldsymbol{x}_t, \boldsymbol{c})|_{\boldsymbol{\theta}=\bar{\boldsymbol{\theta}}} \, \mathrm{d}\boldsymbol{x}_s \right] = \mathbb{E}_{\boldsymbol{c}, \boldsymbol{x}_t} \left[ b(\boldsymbol{c}) \, \nabla_{\boldsymbol{\theta}} \int p_{\boldsymbol{\theta}}(\boldsymbol{x}_s|\boldsymbol{x}_t, \boldsymbol{c}) \mathrm{d}\boldsymbol{x}_s \bigg|_{\boldsymbol{\theta}=\bar{\boldsymbol{\theta}}} \right] = 0. \tag{34}$$

Let $Q_{\bar{\boldsymbol{\theta}}}^{(0,s,t)}$ be the marginal law of $(\boldsymbol{c}, \boldsymbol{x}_0, \boldsymbol{x}_s, \boldsymbol{x}_t)$ under $Q_{\bar{\boldsymbol{\theta}}}^{\text{rev}}$. By the forward-reverse joint-distribution identity used in Theorem 2.1, this same marginal can be written in forward form as

$$Q_{\bar{\boldsymbol{\theta}}}^{(0,s,t)}(\boldsymbol{c}, \boldsymbol{x}_0, \boldsymbol{x}_s, \boldsymbol{x}_t) = p(\boldsymbol{c})\pi_{\bar{\boldsymbol{\theta}}}(\boldsymbol{x}_0|\boldsymbol{c})p(\boldsymbol{x}_s|\boldsymbol{x}_0)p(\boldsymbol{x}_t|\boldsymbol{x}_s), \tag{35}$$

where $\pi_{\bar{\boldsymbol{\theta}}}(\boldsymbol{x}_0|\boldsymbol{c})$ is the terminal distribution induced by the rollout policy. This is a factorization of the same joint law; it does not assert independence between $\boldsymbol{x}_0$ and $\boldsymbol{x}_t$. Using this marginal, Equation (33) becomes

$$\mathbb{E}_{Q_{\bar{\boldsymbol{\theta}}}^{(0,s,t)}} \left[ A(\boldsymbol{x}_0, \boldsymbol{c}) \, \nabla_{\boldsymbol{\theta}} \log p_{\boldsymbol{\theta}}(\boldsymbol{x}_s|\boldsymbol{x}_t, \boldsymbol{c})|_{\boldsymbol{\theta}=\bar{\boldsymbol{\theta}}} \right]. \tag{36}$$

Conditioning on $(\boldsymbol{x}_0, \boldsymbol{c})$ yields

$$\mathbb{E}_{p(\boldsymbol{c})\pi_{\bar{\boldsymbol{\theta}}}(\boldsymbol{x}_0|\boldsymbol{c})} \left[ A(\boldsymbol{x}_0, \boldsymbol{c})\mathbb{E}_{\boldsymbol{x}_s, \boldsymbol{x}_t|\boldsymbol{x}_0} \left[ \nabla_{\boldsymbol{\theta}} \log p_{\boldsymbol{\theta}}(\boldsymbol{x}_s|\boldsymbol{x}_t, \boldsymbol{c})|_{\boldsymbol{\theta}=\bar{\boldsymbol{\theta}}} \right] \right]. \tag{37}$$

From the proof of Theorem 2.1, the per-step Gaussian likelihood can be written, up to a positive scalar and a $\boldsymbol{\theta}$-independent constant, as noisy-target DSM:

$$\log p_{\boldsymbol{\theta}}(\boldsymbol{x}_s|\boldsymbol{x}_t, \boldsymbol{c}) = -\alpha_t \ell_{\boldsymbol{\theta},t}^{\text{noisy}}(\boldsymbol{x}_s, \boldsymbol{x}_t, \boldsymbol{c}) + \kappa_t, \qquad \alpha_t > 0, \tag{38}$$

where $\nabla_{\boldsymbol{\theta}} \kappa_t = 0$ and

$$\ell_{\boldsymbol{\theta},t}^{\text{noisy}}(\boldsymbol{x}_s, \boldsymbol{x}_t, \boldsymbol{c}) := \|\boldsymbol{s}_{\boldsymbol{\theta}}(\boldsymbol{x}_t, t, \boldsymbol{c}) - \nabla_{\boldsymbol{x}_t} \log p(\boldsymbol{x}_t|\boldsymbol{x}_s)\|^2. \tag{39}$$

Therefore,

$$\mathbb{E}_{\boldsymbol{x}_s, \boldsymbol{x}_t|\boldsymbol{x}_0} \left[ \nabla_{\boldsymbol{\theta}} \log p_{\boldsymbol{\theta}}(\boldsymbol{x}_s|\boldsymbol{x}_t, \boldsymbol{c}) \right] = -\alpha_t \nabla_{\boldsymbol{\theta}} \mathbb{E}_{\boldsymbol{x}_s \sim p(\cdot|\boldsymbol{x}_0), \, \boldsymbol{x}_t \sim p(\cdot|\boldsymbol{x}_s)} \left[ \ell_{\boldsymbol{\theta},t}^{\text{noisy}}(\boldsymbol{x}_s, \boldsymbol{x}_t, \boldsymbol{c}) \right]. \tag{40}$$

Now define the clean-target objective

$$\ell_{\boldsymbol{\theta},t}^{\text{clean}}(\boldsymbol{x}_0, \boldsymbol{x}_t, \boldsymbol{c}) := \|\boldsymbol{s}_{\boldsymbol{\theta}}(\boldsymbol{x}_t, t, \boldsymbol{c}) - \nabla_{\boldsymbol{x}_t} \log p(\boldsymbol{x}_t|\boldsymbol{x}_0)\|^2. \tag{41}$$

The conditional version of Lemma 2.2 gives

$$\nabla_{\boldsymbol{\theta}} \mathbb{E}_{\boldsymbol{x}_s \sim p(\cdot|\boldsymbol{x}_0),\, \boldsymbol{x}_t \sim p(\cdot|\boldsymbol{x}_s)} \left[ \ell_{\boldsymbol{\theta},t}^{\mathrm{noisy}}(\boldsymbol{x}_s, \boldsymbol{x}_t, \boldsymbol{c}) \right]$$
$$= \nabla_{\boldsymbol{\theta}} \mathbb{E}_{\boldsymbol{x}_t \sim p(\cdot|\boldsymbol{x}_0)} \left[ \ell_{\boldsymbol{\theta},t}^{\mathrm{clean}}(\boldsymbol{x}_0, \boldsymbol{x}_t, \boldsymbol{c}) \right]. \tag{42}$$

Its proof is the same as the proof of Lemma 2.2, but conditioned on $(\boldsymbol{x}_0, \boldsymbol{c})$ and using the forward identity

$$\int p(\boldsymbol{x}_t|\boldsymbol{x}_s) p(\boldsymbol{x}_s|\boldsymbol{x}_0) \mathrm{d}\boldsymbol{x}_s = p(\boldsymbol{x}_t|\boldsymbol{x}_0). \tag{43}$$

Combining Equations (33), (40) and (42), we obtain

$$\nabla_{\boldsymbol{\theta}} J_t^{\mathrm{DDPO}}(\boldsymbol{\theta}; \bar{\boldsymbol{\theta}}) \big|_{\boldsymbol{\theta}=\bar{\boldsymbol{\theta}}}$$
$$= -\alpha_t \, \nabla_{\boldsymbol{\theta}} \mathbb{E}_{p(\boldsymbol{c})\pi_{\bar{\boldsymbol{\theta}}}(\boldsymbol{x}_0|\boldsymbol{c})p(\boldsymbol{x}_t|\boldsymbol{x}_0)} \left[ A(\boldsymbol{x}_0, \boldsymbol{c}) \ell_{\boldsymbol{\theta},t}^{\mathrm{clean}}(\boldsymbol{x}_0, \boldsymbol{x}_t, \boldsymbol{c}) \right] \big|_{\boldsymbol{\theta}=\bar{\boldsymbol{\theta}}}. \tag{44}$$

Summing adjacent steps over $t$ gives the corresponding local-gradient relation for the full DDPO objective in Equation (5). Thus, after incorporating the terminal advantage, DDPO's local update still maps to an advantage-weighted clean-target DSM/FM gradient; the advantage need not be independent of $\boldsymbol{x}_t$ because the derivation keeps the joint law and conditions on $(\boldsymbol{x}_0, \boldsymbol{c})$.

### C.4. Proof of Theorem 2.3

We first give a more detailed version of Theorem 2.3

**Theorem C.2** (Detailed version of Theorem 2.3 in the main body). *$\boldsymbol{x}_t, \boldsymbol{x}_s$ satisfies*

$$\boldsymbol{x}_t|\boldsymbol{x}_s \sim \mathcal{N}\big(\alpha(t,s)\,\boldsymbol{x}_s,\ \sigma^2(t,s)\,I\big), \qquad \alpha(t,s) = \frac{1-t}{1-s}, \quad \sigma^2(t,s) = t^2 - \frac{(1-t)^2}{(1-s)^2}s^2, \tag{45}$$

*for all $0 \le s < t \le 1$, so that*

$$\sigma^2(t,s) + \alpha^2(t,s)\,s^2 \ = \ t^2. \tag{46}$$

*Let $d$ be the data dimension. Then, for the two conditional scores*

$$\nabla_{\boldsymbol{x}_t} \log p(\boldsymbol{x}_t|\boldsymbol{x}_s), \qquad \nabla_{\boldsymbol{x}_t} \log p(\boldsymbol{x}_t|\boldsymbol{x}_0),$$

*and for any fixed $\boldsymbol{x}_t$ and any $s \in [0,t)$, each is an unbiased estimator of the true score $\nabla_{\boldsymbol{x}_t} \log p(\boldsymbol{x}_t)$:*

$$\mathbb{E}[\nabla_{\boldsymbol{x}_t} \log p(\boldsymbol{x}_t|\boldsymbol{x}_s) \,|\, \boldsymbol{x}_t] = \mathbb{E}[\nabla_{\boldsymbol{x}_t} \log p(\boldsymbol{x}_t|\boldsymbol{x}_0) \,|\, \boldsymbol{x}_t] = \nabla_{\boldsymbol{x}_t} \log p(\boldsymbol{x}_t).$$

*Moreover, their conditional covariances satisfy*

$$\mathrm{Cov}(\nabla_{\boldsymbol{x}_t} \log p(\boldsymbol{x}_t|\boldsymbol{x}_s) \,|\, \boldsymbol{x}_t) = \mathrm{Cov}(\nabla_{\boldsymbol{x}_t} \log p(\boldsymbol{x}_t|\boldsymbol{x}_0) \,|\, \boldsymbol{x}_t) + \kappa(s,t)\,I \ \succeq \ \mathrm{Cov}(\nabla_{\boldsymbol{x}_t} \log p(\boldsymbol{x}_t|\boldsymbol{x}_0) \,|\, \boldsymbol{x}_t), \tag{47}$$

*where*

$$\kappa(s,t) \ = \ \frac{(1-t)^2\, s^2}{t^2\big(t^2(1-s)^2 - s^2(1-t)^2\big)}. \tag{48}$$

*In particular,*

$$\mathrm{Tr}\,\mathrm{Cov}(\nabla_{\boldsymbol{x}_t} \log p(\boldsymbol{x}_t|\boldsymbol{x}_s) \,|\, \boldsymbol{x}_t) = \mathrm{Tr}\,\mathrm{Cov}(\nabla_{\boldsymbol{x}_t} \log p(\boldsymbol{x}_t|\boldsymbol{x}_0) \,|\, \boldsymbol{x}_t) + d\,\kappa(s,t).$$

*The map $s \mapsto \kappa(s,t)$ is strictly increasing on $[0,t)$, with $\kappa(0,t) = 0$ and $\kappa(s,t) \to \infty$ as $s \uparrow t$.*

*Moreover, for any predictor $a = a(\boldsymbol{x}_t, t) \in \mathbb{R}^d$ (e.g. $a = \boldsymbol{s}_\theta(\boldsymbol{x}_t, t)$):*

$$\mathbb{E}\left[\|a - \nabla_{\boldsymbol{x}_t} \log p(\boldsymbol{x}_t|\boldsymbol{x}_s)\|^2 \,\Big|\, \boldsymbol{x}_t\right] = \mathbb{E}\left[\|a - \nabla_{\boldsymbol{x}_t} \log p(\boldsymbol{x}_t|\boldsymbol{x}_0)\|^2 \,\Big|\, \boldsymbol{x}_t\right] + d\,\kappa(s,t), \tag{49}$$

$$\mathrm{Var}\left(\|a - \nabla_{\boldsymbol{x}_t} \log p(\boldsymbol{x}_t|\boldsymbol{x}_s)\|^2 \,\Big|\, \boldsymbol{x}_t\right) = \mathrm{Var}\left(\|a - \nabla_{\boldsymbol{x}_t} \log p(\boldsymbol{x}_t|\boldsymbol{x}_0)\|^2 \,\Big|\, \boldsymbol{x}_t\right) + 2d\,\kappa(s,t)^2$$
$$+ 4\,\kappa(s,t)\,\mathbb{E}\left[\|a - \nabla_{\boldsymbol{x}_t} \log p(\boldsymbol{x}_t|\boldsymbol{x}_0)\|^2 \,\Big|\, \boldsymbol{x}_t\right]. \tag{50}$$

*Hence both the conditional risk and the conditional objective variance are minimized at $s = 0$ and strictly increase with $s \in (0, t)$.*

*Proof.* Under Equation (45),

$$\nabla_{\boldsymbol{x}_t} \log p(\boldsymbol{x}_t|\boldsymbol{x}_s) = -\frac{1}{\sigma^2(t,s)}\left(\boldsymbol{x}_t - \alpha(t,s)\,\boldsymbol{x}_s\right), \qquad \nabla_{\boldsymbol{x}_t} \log p(\boldsymbol{x}_t|\boldsymbol{x}_0) = -\frac{1}{t^2}\left(\boldsymbol{x}_t - (1-t)\,\boldsymbol{x}_0\right). \tag{51}$$

**Auxiliary Lemma 1**   For fixed $\boldsymbol{x}_t$ and any $s \in [0,t)$, there exists a zero-mean Gaussian vector $\boldsymbol{\eta}$ such that

$$\nabla_{\boldsymbol{x}_t} \log p(\boldsymbol{x}_t|\boldsymbol{x}_s) = \nabla_{\boldsymbol{x}_t} \log p(\boldsymbol{x}_t|\boldsymbol{x}_0) + \boldsymbol{\eta}, \qquad \boldsymbol{\eta} \perp (\boldsymbol{x}_0|\boldsymbol{x}_t), \qquad \boldsymbol{\eta} \sim \mathcal{N}\big(\boldsymbol{0},\ \kappa(s,t)\,I\big), \tag{52}$$

with $\kappa(s,t)$ given by Equation (48).

*Proof of Aux. Lemma 1.* Conditioned on $(\boldsymbol{x}_t, \boldsymbol{x}_0)$, the posterior of $\boldsymbol{x}_s$ is Gaussian with mean $\frac{\sigma^2(t,s)(1-s)\boldsymbol{x}_0 + \alpha(t,s)s^2\boldsymbol{x}_t}{t^2}$ and covariance $\frac{s^2\sigma^2(t,s)}{t^2}$ Using equation 51, define

$$\boldsymbol{\eta} := \nabla_{\boldsymbol{x}_t} \log p(\boldsymbol{x}_t|\boldsymbol{x}_s) - \nabla_{\boldsymbol{x}_t} \log p(\boldsymbol{x}_t|\boldsymbol{x}_0) = \frac{\alpha(t,s)}{\sigma^2(t,s)}\left(\boldsymbol{x}_s - \mathbb{E}[\boldsymbol{x}_s|\boldsymbol{x}_t, \boldsymbol{x}_0]\right).$$

Then $\boldsymbol{\eta}|(\boldsymbol{x}_t, \boldsymbol{x}_0)$ is zero-mean Gaussian with covariance

$$\frac{\alpha^2}{\sigma^4}\frac{s^2\sigma^2}{t^2}I = \frac{\alpha^2 s^2}{\sigma^2 t^2}\,I = \kappa(s,t)\,I.$$

The residual $\boldsymbol{x}_s - \mathbb{E}[\boldsymbol{x}_s|\boldsymbol{x}_t, \boldsymbol{x}_0]$ is independent of $(\boldsymbol{x}_t, \boldsymbol{x}_0)$, hence $\boldsymbol{\eta} \perp (\boldsymbol{x}_0|\boldsymbol{x}_t)$.   $\square$

By the score identity, $\mathbb{E}[\nabla_{\boldsymbol{x}_t} \log p(\boldsymbol{x}_t|\boldsymbol{x}_s)|\boldsymbol{x}_t] = \nabla_{\boldsymbol{x}_t} \log p(\boldsymbol{x}_t)$ for every $s \in [0,t)$; in particular this equals $\mathbb{E}[\nabla_{\boldsymbol{x}_t} \log p(\boldsymbol{x}_t|\boldsymbol{x}_0)|\boldsymbol{x}_t]$. From equation 52 and $\boldsymbol{\eta} \perp (\boldsymbol{x}_0|\boldsymbol{x}_t)$,

$$\begin{aligned}
\mathrm{Cov}(\nabla_{\boldsymbol{x}_t} \log p(\boldsymbol{x}_t|\boldsymbol{x}_s)\,|\,\boldsymbol{x}_t) &= \mathrm{Cov}(\nabla_{\boldsymbol{x}_t} \log p(\boldsymbol{x}_t|\boldsymbol{x}_0)\,|\,\boldsymbol{x}_t) + \mathrm{Cov}(\boldsymbol{\eta}|\boldsymbol{x}_t)\\
&= \mathrm{Cov}(\nabla_{\boldsymbol{x}_t} \log p(\boldsymbol{x}_t|\boldsymbol{x}_0)\,|\,\boldsymbol{x}_t) + \kappa(s,t)\,I,
\end{aligned}$$

which proves Theorem 2.3 and the trace statement.

**Auxiliary Lemma 2 (Monotonicity of $\kappa$).**   For fixed $t \in (0,1]$, $s \mapsto \kappa(s,t)$ in Equation (48) is strictly increasing on $[0,t)$, with $\kappa(0,t) = 0$ and $\kappa(s,t) \to \infty$ as $s \uparrow t$.

*Proof of Aux. Lemma 2.* Using Equation (46), write $\kappa(s,t) = \frac{q(s)}{t^2\left(t^2 - q(s)\right)}$ with $q(s) := \alpha^2(t,s)s^2 = \frac{(1-t)^2 s^2}{(1-s)^2}$. On $[0,t)$, $q(s)$ is strictly increasing and takes values in $[0,t^2)$, while $q \mapsto \frac{q}{t^2(t^2-q)}$ is strictly increasing on $[0,t^2)$. Limits follow from $q(0) = 0$ and $q(s) \uparrow t^2$ as $s \uparrow t$.   $\square$

For any $a$,

$$\mathbb{E}\Big[\big\|a - \nabla_{\boldsymbol{x}_t} \log p(\boldsymbol{x}_t|\boldsymbol{x}_s)\big\|^2 \,\Big|\, \boldsymbol{x}_t\Big] = \Big\|a - \mathbb{E}[\nabla_{\boldsymbol{x}_t} \log p(\boldsymbol{x}_t|\boldsymbol{x}_s)|\boldsymbol{x}_t]\Big\|^2 + \mathrm{Tr}\,\mathrm{Cov}(\nabla_{\boldsymbol{x}_t} \log p(\boldsymbol{x}_t|\boldsymbol{x}_s)\,|\,\boldsymbol{x}_t).$$

Insert unbiasedness and Equation (47), then subtract the same identity with $s = 0$ to obtain the increment $d\,\kappa(s,t)$.

To prove Equation (50), let $\boldsymbol{\delta}(\boldsymbol{x}_0) := a - \nabla_{\boldsymbol{x}_t} \log p(\boldsymbol{x}_t|\boldsymbol{x}_0)$. From Equation (52),

$$\big\|a - \nabla_{\boldsymbol{x}_t} \log p(\boldsymbol{x}_t|\boldsymbol{x}_s)\big\|^2 = \|\boldsymbol{\delta} - \boldsymbol{\eta}\|^2 = \|\boldsymbol{\delta}\|^2 + \|\boldsymbol{\eta}\|^2 - 2\langle\boldsymbol{\delta}, \boldsymbol{\eta}\rangle.$$

Conditioned on $(\boldsymbol{x}_t, \boldsymbol{x}_0)$, $\boldsymbol{\eta} \sim \mathcal{N}(0, \kappa I)$ is independent of $\boldsymbol{\delta}$. Gaussian moment identities yield, coordinatewise,

$$\mathbb{E}[\|\boldsymbol{\eta}\|^2|\boldsymbol{x}_t, \boldsymbol{x}_0] = d\,\kappa, \quad \mathrm{Var}(\|\boldsymbol{\eta}\|^2|\boldsymbol{x}_t, \boldsymbol{x}_0) = 2d\,\kappa^2,$$
$$\mathrm{Var}(\langle\boldsymbol{\delta}, \boldsymbol{\eta}\rangle|\boldsymbol{x}_t, \boldsymbol{x}_0) = \kappa\,\|\boldsymbol{\delta}\|^2, \quad \mathrm{Cov}(\|\boldsymbol{\eta}\|^2, \langle\boldsymbol{\delta}, \boldsymbol{\eta}\rangle|\boldsymbol{x}_t, \boldsymbol{x}_0) = 0.$$

Therefore,

$$\mathrm{Var}\Big(\big\|a - \nabla_{\boldsymbol{x}_t} \log p(\boldsymbol{x}_t|\boldsymbol{x}_s)\big\|^2 \,\Big|\, \boldsymbol{x}_t, \boldsymbol{x}_0\Big) = \mathrm{Var}\big(\|\boldsymbol{\eta}\|^2 - 2\langle\boldsymbol{\delta}, \boldsymbol{\eta}\rangle|\boldsymbol{x}_t, \boldsymbol{x}_0\big) = 2d\,\kappa^2 + 4\kappa\,\|\boldsymbol{\delta}\|^2.$$

Law of total variance yields Equation (50). Monotonicity in $s$ follows from Aux. Lemma 2; strictness for $s \in (0,t)$ follows since $\kappa(s,t) > 0$.   $\square$

## C.5. Deviation of Equation (12)

In this section, we provide the detailed derivation of the noisy Denoising Score Matching (DSM) objective under the EDM parameterization (Equation 12).

**1. Conditional Distribution** $p(\boldsymbol{x}_t|\boldsymbol{x}_s)$**.** The EDM noise schedule is defined as $\boldsymbol{x}_t = \boldsymbol{x}_0 + t\boldsymbol{\epsilon}$, where $\boldsymbol{\epsilon} \sim \mathcal{N}(\boldsymbol{0}, \boldsymbol{I})$. For $0 \leq s < t$, the transition from $\boldsymbol{x}_s$ to $\boldsymbol{x}_t$ follows a Gaussian distribution due to the properties of the equivalent variance-exploding SDE. The conditional distribution is given by:

$$p(\boldsymbol{x}_t|\boldsymbol{x}_s) = \mathcal{N}(\boldsymbol{x}_t; \boldsymbol{x}_s, (t^2 - s^2)\boldsymbol{I}). \tag{53}$$

**2. Conditional Score** $\nabla_{\boldsymbol{x}_t} \log p(\boldsymbol{x}_t|\boldsymbol{x}_s)$**.** The score function corresponding to this conditional distribution is the gradient of the log-likelihood with respect to $\boldsymbol{x}_t$:

$$\nabla_{\boldsymbol{x}_t} \log p(\boldsymbol{x}_t|\boldsymbol{x}_s) = \nabla_{\boldsymbol{x}_t} \left( -\frac{1}{2(t^2 - s^2)} \|\boldsymbol{x}_t - \boldsymbol{x}_s\|^2 + \text{const} \right)$$
$$= -\frac{\boldsymbol{x}_t - \boldsymbol{x}_s}{t^2 - s^2}. \tag{54}$$

**3. Relation between Denoiser** $\boldsymbol{D}_{\boldsymbol{\theta}}$ **and Score.** In the EDM framework, the model is parameterized as a denoiser $\boldsymbol{D}_{\boldsymbol{\theta}}(\boldsymbol{x}_t, t)$ trained to predict the clean data $\boldsymbol{x}_0$. The relationship between the optimal denoiser and the score function is (Tweedie's formula):

$$\boldsymbol{D}_{\boldsymbol{\theta}}(\boldsymbol{x}_t, t) = \boldsymbol{x}_t + t^2 \nabla_{\boldsymbol{x}_t} \log p(\boldsymbol{x}_t). \tag{55}$$

In the context of Noisy DSM, we substitute the true score $\nabla_{\boldsymbol{x}_t} \log p(\boldsymbol{x}_t)$ with the conditional score $\nabla_{\boldsymbol{x}_t} \log p(\boldsymbol{x}_t|\boldsymbol{x}_s)$ as the regression target.

**4. Target for** $\boldsymbol{D}_{\boldsymbol{\theta}}$**.** By substituting the conditional score into the denoiser equation, we obtain the training target for $\boldsymbol{D}_{\boldsymbol{\theta}}(\boldsymbol{x}_t, t)$:

$$\text{Target}(\boldsymbol{x}_t, \boldsymbol{x}_s) = \boldsymbol{x}_t + t^2 \left( \nabla_{\boldsymbol{x}_t} \log p(\boldsymbol{x}_t|\boldsymbol{x}_s) \right)$$
$$= \boldsymbol{x}_t + t^2 \left( -\frac{\boldsymbol{x}_t - \boldsymbol{x}_s}{t^2 - s^2} \right)$$
$$= \boldsymbol{x}_t - \frac{t^2}{t^2 - s^2} (\boldsymbol{x}_t - \boldsymbol{x}_s). \tag{56}$$

Therefore, the Noisy DSM loss function in terms of the denoiser $\boldsymbol{D}_{\boldsymbol{\theta}}$ is:

$$\mathcal{L}_{\text{Noisy-DSM}} = \mathbb{E}_{\boldsymbol{x}_s, \boldsymbol{x}_t} \left[ w(t) \left\| \boldsymbol{D}_{\boldsymbol{\theta}}(\boldsymbol{x}_t, t) - \left( \boldsymbol{x}_t - \frac{t^2}{t^2 - s^2} (\boldsymbol{x}_t - \boldsymbol{x}_s) \right) \right\|^2 \right], \tag{57}$$

which matches Equation 12.

## D. Detailed description of DDPO

Denoising diffusion policy optimization (DDPO) (Black et al., 2023) map the iterative denoising procedure to the following MDP:

**Problem Setup of Denoising Diffusion Policy Optimization**

$$\text{State:} \quad \boldsymbol{s}_t = (\boldsymbol{c}, t, \boldsymbol{x}_t) \qquad \text{Action:} \quad \boldsymbol{a}_t = \boldsymbol{x}_{t-1} \qquad \text{Policy:} \quad \pi(\boldsymbol{a}_t|\boldsymbol{s}_t) = p_{\boldsymbol{\theta}}(\boldsymbol{x}_{t-1}|\boldsymbol{x}_t, \boldsymbol{c})$$

To calculate the reverse transition probability $p_{\boldsymbol{\theta}}(\boldsymbol{x}_{t-1}|\boldsymbol{x}_t, \boldsymbol{c})$, we first consider the sampling process of diffusion or flow matching models involves solving the reverse time diffusion SDE (Song et al., 2020), where $\boldsymbol{w}_t$ represents the Wiener process:

$$\mathrm{d}\boldsymbol{x}_t = \left[\boldsymbol{v}_{\boldsymbol{\theta}}(\boldsymbol{x}_t, t) + \frac{1}{1-t}\left(\boldsymbol{x}_t + (1-t)\boldsymbol{v}_{\boldsymbol{\theta}}(\boldsymbol{x}_t, t)\right)\right]\mathrm{d}t + \sqrt{\frac{2t}{1-t}}\mathrm{d}\boldsymbol{w}_t,$$

However, the exact transition probability $p_{\boldsymbol{\theta}}(\boldsymbol{x}_{t-1}|\boldsymbol{x}_t, \boldsymbol{c})$ is analytically intractable. To address this, DDPO implements the Euler–Maruyama discretization of the reverse SDE:

$$\boldsymbol{x}_{t-\Delta t} = \boldsymbol{x}_t - \left[\boldsymbol{v}_{\boldsymbol{\theta}}(\boldsymbol{x}_t, t) + \frac{1}{1-t}\left(\boldsymbol{x}_t + (1-t)\boldsymbol{v}_{\boldsymbol{\theta}}(\boldsymbol{x}_t, t)\right)\right]\Delta t + \sqrt{\frac{2t}{1-t}}\sqrt{\Delta t}\boldsymbol{\epsilon},$$

Then under the discretization, the transition probability $p_{\boldsymbol{\theta}}(\boldsymbol{x}_{t-1}|\boldsymbol{x}_t, \boldsymbol{c}) \sim \mathcal{N}(\boldsymbol{\mu}_{\boldsymbol{\theta}}, \sigma^2 I)$ becomes an isotropic Gaussian with tractable mean $\boldsymbol{\mu}_{\boldsymbol{\theta}}$ and variance $\sigma^2$:

$$\boldsymbol{\mu}_{\boldsymbol{\theta}} = \boldsymbol{x}_t - \left[\boldsymbol{v}_{\boldsymbol{\theta}}(\boldsymbol{x}_t, t) + \frac{1}{1-t}\left(\boldsymbol{x}_t + (1-t)\boldsymbol{v}_{\boldsymbol{\theta}}(\boldsymbol{x}_t, t)\right)\right]\Delta t, \quad \sigma^2 = \frac{2t}{1-t}\Delta t,$$

yielding the per-step log-likelihood $\log p_{\boldsymbol{\theta}}(\boldsymbol{x}_{t-\Delta t}|\boldsymbol{x}_t)$:

$$-\left\|\boldsymbol{x}_{t-\Delta t} - \left(\boldsymbol{x}_t - \left[\boldsymbol{v}_{\boldsymbol{\theta}}(\boldsymbol{x}_t, t) + \frac{1}{1-t}\left(\boldsymbol{x}_t + (1-t)\boldsymbol{v}_{\boldsymbol{\theta}}(\boldsymbol{x}_t, t)\right)\right]\Delta t\right)\right\|^2 / (\frac{4t}{1-t}\Delta t) + \text{const.}$$

Then the DDPO's objective is

$$\mathbb{E}_{p_{\boldsymbol{\theta}_{\text{old}}}}\left[\sum_{t=0}^{T}\frac{p_{\boldsymbol{\theta}}(\boldsymbol{x}_{t-1}|\boldsymbol{x}_t, \boldsymbol{c})}{p_{\boldsymbol{\theta}_{\text{old}}}(\boldsymbol{x}_{t-1}|\boldsymbol{x}_t, \boldsymbol{c})}\left(r(\boldsymbol{x}_0, \boldsymbol{c}) - b(\boldsymbol{c})\right)\right]. \tag{58}$$

where $b(\boldsymbol{c})$ denotes an arbitrary baseline independent of $\boldsymbol{x}_0$. This baseline does not change the expected gradient but can significantly reduce its variance (Greensmith et al., 2004). The baseline can be computed in various ways (e.g., PPO (Schulman et al., 2017) or GRPO (Shao et al., 2024)) and does not affect the correctness of the objective.

# E. Related work

## E.1. Reward Feedback Learning

Imagereward (Xu et al., 2023) proposes Reward Feedback Learning (ReFL), which directly maximizes the reward of an approximately one-step predicted image via gradient backpropagation. To mitigate the approximation error of the one-step prediction, DRaFT (Clark et al., 2023) maximizes the reward of the final multi-step sampled images. While this incurs a massive memory cost, DRaFT reduces it through backpropagation truncation and gradient checkpointing. Viewing the problem from a continuous-time perspective, Adjoint Matching (Domingo-Enrich et al., 2024) applies the adjoint method (Pontryagin, 2018) for memory-efficient gradient computation. There are also trials of applying Reward Feedback Learning on few-step distilled models (Kim et al., 2024; Li et al., 2024; Luo et al., 2024; 2025).

However, the Reward Feedback Learning method has some inevitable drawbacks. First, the reward function must be differentiable, which prevents the use of rule-based or binary rewards, such as those from detection models like GenEval (Ghosh et al., 2023) or OCR. Second, modern generative models operate in a highly compressed VAE latent space. Backpropagating gradients from pixel-level rewards through the VAE decoder adds non-negligible memory and computation costs. Third, although first-order optimization is generally more efficient than zero-order optimization (Nocedal & Wright, 2006), it carries a higher risk of reward hacking, where the model exploits the reward function in unintended ways.

## E.2. Denoising Diffusion Policy Optimization

DDPO (Black et al., 2023) and DPOK (Fan et al., 2023) frame the reinforcement learning problem as a multi-step decision-making problem. They consider an Euler-Maruyama discretization of the reverse process Equation (1), leading to a tractable Gaussian likelihood at each step. DeepSeekMath (Shao et al., 2024) proposed Group Relative Policy Optimization (GRPO) by substituting the baseline from an additional Value model in PPO (Schulman et al., 2017) to a group relative mean of reward. Flow-GRPO (Liu et al., 2025a) and DanceGRPO (Xue et al., 2025) combine the DDPO formulation and use GRPO

*Table 4.* Performance comparison on GenEval with early experiment configurations. Speed-up is relative to Flow-GRPO baseline.

| Model | GenEval | | | | | | | |
| --- | --- | --- | --- | --- | --- | --- | --- | --- |
| | Single Obj. | Two Obj. | Counting | Color | Position | Attr | Overall | Speed-up |
| DALLE-3 | 0.96 | 0.87 | 0.47 | 0.83 | 0.43 | 0.45 | 0.67 | – |
| GPT-4o | 0.99 | 0.92 | 0.85 | 0.92 | 0.75 | 0.61 | 0.84 | – |
| SD-XL | 0.98 | 0.74 | 0.39 | 0.85 | 0.15 | 0.23 | 0.55 | – |
| FLUX.1 Dev | 0.98 | 0.81 | 0.74 | 0.79 | 0.22 | 0.45 | 0.66 | – |
| SD3.5L | 0.98 | 0.89 | 0.73 | 0.83 | 0.34 | 0.47 | 0.71 | – |
| SD3.5M | 0.98 | 0.78 | 0.50 | 0.81 | 0.24 | 0.52 | 0.63 | – |
| Flow-GRPO | 1.00 | 0.99 | 0.95 | 0.92 | 0.99 | 0.86 | **0.95** | $1\times$ |
| AWM (Ours) | 1.00 | 0.99 | 0.95 | 0.93 | 0.98 | 0.83 | **0.95** | $\mathbf{8.02\times}$ |

*Table 5.* Performance comparison on OCR/PickScore for SD3.5M and FLUX with early experiment configurations. Hours is the total GPU hours; Speed-up is relative to Flow-GRPO baseline. [†] indicates longer training time.

| Method | SD3.5M | | | | FLUX | | | |
| --- | --- | --- | --- | --- | --- | --- | --- | --- |
| | OCR | | PickScore | | OCR | | PickScore | |
| | Acc | Hours | Score | Hours | Acc | Hours | Score | GPU hours |
| Base model | 0.59 | – | 21.72 | – | 0.59 | – | 22.20 | – |
| FlowGRPO | 0.89 | 415.9 $(1\times)$ | 23.01 | 956.1 $(1\times)$ | 0.95 | 343.6 $(1\times)$ | 23.08 | 339.2 $(1\times)$ |
| AWM (Ours) | 0.89 | 17.6 $(23.59\times)$ | 23.02 | 91.1 $(10.49\times)$ | 0.95 | 40.3 $(8.53\times)$ | 23.08 | 49.8 $(6.82\times)$ |
| AWM (Ours)[†] | 0.95 (+6.74%) | 79.0 | 23.25 (+0.99%) | 205.0 | 0.99 (+4.21%) | 147.0 | 23.18 (+0.43%) | 78.0 |

as its RL algorithm. TempFlow (He et al., 2025) introduces a trajectory branching mechanism that provides process rewards by concentrating stochasticity at designated branching points. MixGRPO (Li et al., 2025a) leverages the flexibility of mixed sampling strategies through the integration of ODE and SDE sampling. BranchGRPO (Li et al., 2025b) introduces a branching scheme that amortizes rollout cost through shared prefixes while preserving exploration diversity.

### E.3. Reward Weighted Regression

Lee et al. (2023) proposes to finetune a text-to-image model by maximizing an offline reward-weighted denoising loss. Fan et al. (2025) considers an online version of reward-weighted denoising loss with a Wasserstein-2 regularization. Concurrent to ours, FMPG (McAllister et al., 2025) proposes using the ELBO as a proxy for policy likelihood. Unlike our approach, they do not articulate the connection to DDPO, and they do not present experiments on text-to-image diffusion.

Although the name is reminiscent of reward-weighted or advantage-weighted regression, AWM differs from classical RWR/AWR objectives (Peters & Schaal, 2007; Peng et al., 2019) at the level of the weights. RWR/AWR fit a policy by maximum-likelihood regression with nonnegative exponential weights, e.g., $\exp(r/\beta)$ or $\exp(A/\beta)$. Thus, even low-advantage samples still contribute a positive regression weight, only with a smaller magnitude. In contrast, AWM is derived from a policy-gradient objective and keeps the advantage as a signed, linear weight on the clean-target score-matching loss:

$$\nabla_{\boldsymbol{\theta}} J_{\mathrm{AWM}} \propto -\nabla_{\boldsymbol{\theta}} \mathbb{E}\big[A(\boldsymbol{x}_0, \boldsymbol{c})\,\ell_{\boldsymbol{\theta}}^{\mathrm{clean}}(\boldsymbol{x}_0, \boldsymbol{x}_t, \boldsymbol{c})\big].$$

Consequently, positive-advantage samples pull the model toward their clean targets, while negative-advantage samples reverse the gradient direction and suppress matching to low-reward samples. This signed policy-gradient weighting, together with the pretraining-compatible DSM/FM loss, is the key distinction from density-regression-style RWR/AWR.

### E.4. Continuous-time RL and Control for Diffusion

Gao et al. (2025) cast reward-directed diffusion as an entropy-regularized continuous-time RL problem with a running penalty that pushes the denoising dynamics toward the (unknown) true score. They avoid relying on a pretrained score by using a simple ratio estimator to obtain noisy score observations, and solve the problem with a Gaussian-policy q-learning algorithm.

Zhao et al. (2025) treat the score itself as the *action* in a continuous-time stochastic control/RL objective, optimizing terminal reward with a time-integrated KL regularizer between the fine-tuned and pretrained distributions; they derive continuous-time policy-gradient results and report robustness to time discretization.

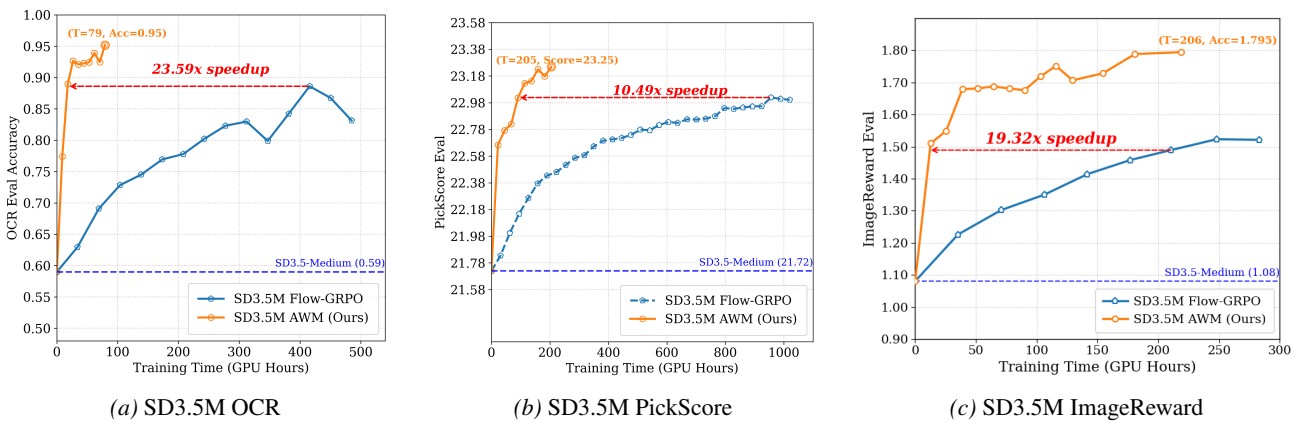

*(a)* SD3.5M OCR        *(b)* SD3.5M PickScore        *(c)* SD3.5M ImageReward

*Figure 9.* **OCR, PickScore, and ImageReward training efficiency with early experiment configurations.** Metric vs. GPU hours for SD3.5M. AWM (ours) exceeds Flow-GRPO with far less compute.

## F. Experiment Settings and Additional Experiments

### F.1. Experiment Settings

We use a group size $G = 24$, and LoRA (Hu et al., 2022) with $\alpha = 64$ and $r = 32$ for SD3.5M and LoRA with $\alpha = 128$ and $r = 64$ for FLUX. The learning rate is set to a constant $3e - 4$.

In our early experiment, we use weighting $w(t) = 1$ in Equation (15), and adopt Euler-Maruyama sampler with noise level $\eta = 0.7$ with 20 timesteps and a guidance scale of cfg $= 4.5$ for sampling. We use 4 Monte Carlo samples to estimate the ELBO, which also corresponds to the number of training timesteps. We set the reference model to the base model, and the KL ratio $\beta$ is set to 0.4 for GenEval and OCR, and 0.01 for PickScore. We split each generated batch into two equal parts: one half is used for an on-policy update, while the other half is used for a single-step off-policy update with importance sampling ratios (the configurations for sampling is exactly the same with original Flow-GRPO paper).

In later experiments, we observe improved performance when using $w(t) = t$ in Equation (15), together with SA-Solver (Xue et al., 2023) with noise level $\eta = 0.4$, 14 timesteps, and a guidance scale of cfg $= 1.0$. We use 6 Monte Carlo samples to estimate the ELBO. In this setting, we set the reference model to an Exponential Moving Average of the running model, with the EMA rate set as $\eta = \min \{0.001t, 0.3\}$, where $t$ is the iter number. The KL ratio $\beta$ is set to 1 for GenEval, OCR, and PickScore. We maintain a purely on-policy training regime, where each generated batch is used for a single gradient update without off-policy reuse.

Unless otherwise specified as in this subsection in appendix, all experiments reported in the main paper use the latter configuration.

### F.2. Additional Experiments

We provide additional experimental results obtained under the earlier configuration. These results include full training curves and final performance metrics across all benchmarks, and serve to complement the ablation studies by illustrating the behavior of our method under an alternative training setting.

Table 4 reports the detailed GenEval results for SD3.5M, including per-category scores and overall performance. Table 5 presents additional comparisons on OCR and PickScore for SD3.5M and FLUX, including total GPU hours and relative speed-ups. These results further illustrate the efficiency gains of AWM across different backbones and evaluation metrics. Figure 9 shows training curves of OCR, PickScore, and ImageReward as a function of GPU hours on SD3.5M. Compared to Flow-GRPO, AWM reaches higher evaluation scores with substantially fewer compute resources across all metrics.

### F.3. Target Variance

To directly validate the variance prediction in Theorem 2.3, we measure the conditional target variance of the clean-target and noisy-target DSM formulations under the EDM noise schedule on CIFAR-10 and ImageNet-64. As shown in Table 6, the noisy target consistently has much larger variance across all noise levels and both datasets, matching the theoretical

*Table 6.* **Target variance.** Conditional target variance for clean-target DSM and noisy-target DSM under the EDM noise schedule. The noisy-target formulation, which mirrors the DDPO target, has substantially larger variance across all measured noise levels.

| | $\sigma = 0.002$ | $\sigma = 0.006$ | $\sigma = 0.019$ | $\sigma = 0.057$ | $\sigma = 0.173$ | $\sigma = 0.529$ | $\sigma = 1.61$ | $\sigma = 4.92$ | $\sigma = 15.0$ | $\sigma = 80.0$ |
|---|---|---|---|---|---|---|---|---|---|---|
| **CIFAR-10 clean** | 0.00e+00 | 0.00e+00 | 0.00e+00 | 0.00e+00 | 0.00e+00 | 0.00e+00 | 4.12e-06 | 126 | 597 | 694 |
| **CIFAR-10 noisy** | 0.02 | 0.15 | 1.37 | 12.74 | 119 | 1104 | 10274 | 95759 | 890747 | 2.53e+07 |
| **ImageNet-64 clean** | 0.00e+00 | 0.00e+00 | 0.00e+00 | 0.00e+00 | 0.00e+00 | 0.00e+00 | 0.00e+00 | 0.03 | 1181 | 2665 |
| **ImageNet-64 noisy** | 0.06 | 0.59 | 5.48 | 50.96 | 474 | 4415 | 41097 | 382533 | 3.56e+06 | 1.01e+08 |

covariance ordering.

## G. Additional Samples

As shown in Figure 10, even after 600 RL optimization steps, our method preserves broad mode coverage: four samples generated from identical prompts but different seeds remain visually diverse rather than collapsing into a single mode.

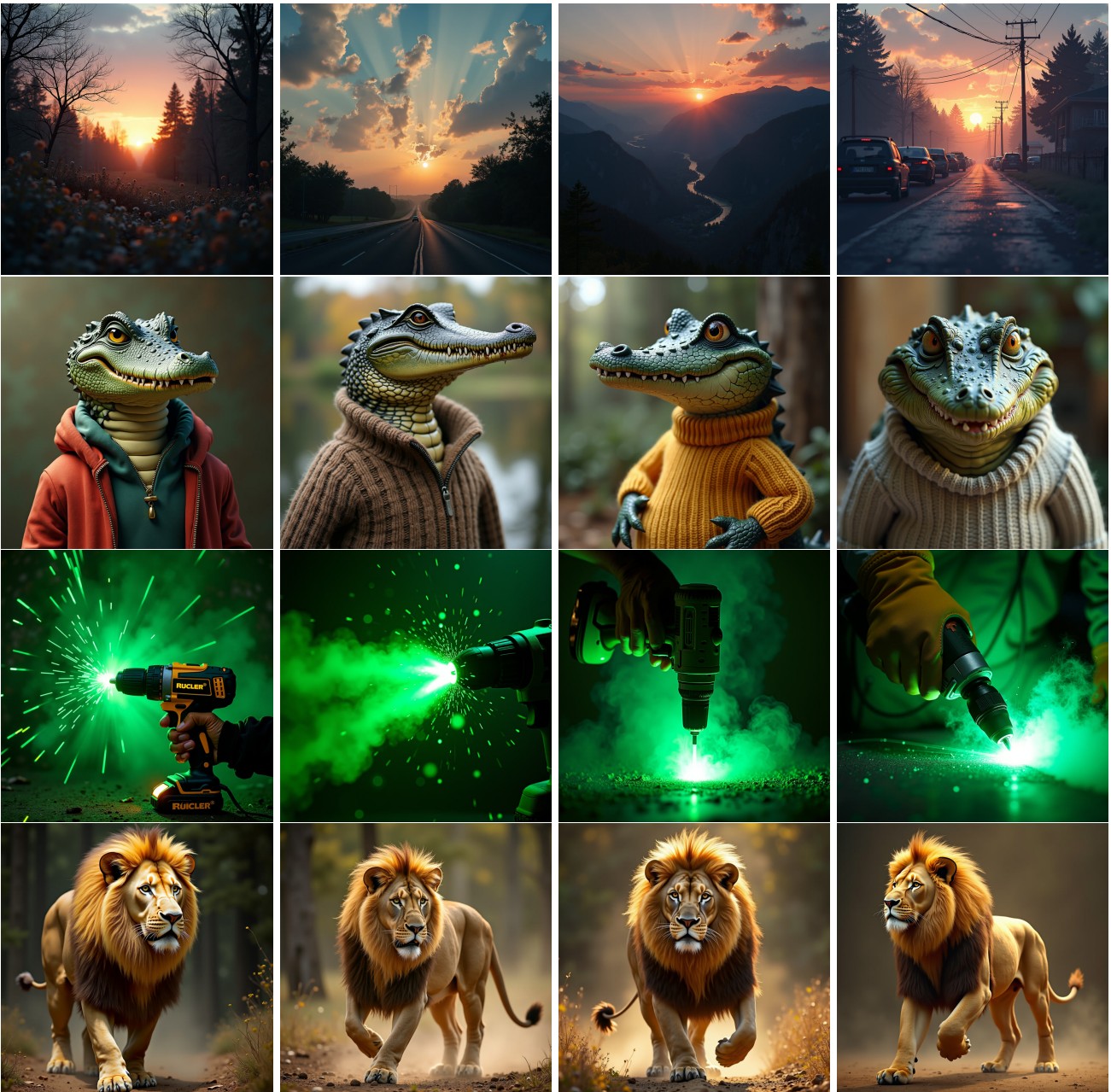

*Figure 10.* **FLUX after 600 RL steps**. Using the same prompt, four different seeds (initial noise) are sampled for each row. Our method maintains strong sample diversity. **Row 1**: "Amid the soft glow of twilight." **Row 2**: "Crocodile in a sweater." **Row 3**: "A vivid nuclear green electric drill in action." **Row 4**: "A regal lion."

