# OpenReview forum: "Advantage Weighted Matching: Aligning RL with Pretraining in Diffusion Models"
_ICML.cc/2026/Conference — ICML 2026 regular_

### Official Review · Reviewer_i7aJ · 2026-03-12

**Soundness:** 2
**Presentation:** 1
**Significance:** 3
**Originality:** 2
**Overall Recommendation:** 3
**Confidence:** 4

**Summary:**

This paper focuses on improving reinforcement learning based fine-tuning for diffusion models. The authors first analyze the existing Denoising Diffusion Policy Optimization (DDPO) framework and show that it can be interpreted as an implicit form of score matching with a noisy target. They argue that this formulation leads to a higher-variance target estimator, which in turn slows down convergence during training. To address this issue, the paper proposes a new method called Advantage Weighted Matching (AWM). Unlike prior approaches that perform score matching using consecutive noisy states $(x_t, x_s)$ in the reverse diffusion process, AWM performs score matching using the clean sample $x_0$ and its forward-diffused state $x_t$. By relying on clean-data targets and weighting the matching objective with advantages derived from rewards, the method aims to reduce variance and improve training efficiency. They evaluate the proposed approach on modern diffusion backbones such as Stable Diffusion 3.5 and FLUX, and demonstrate that AWM achieves improved performance across several benchmarks while requiring significantly fewer training iterations compared to existing methods.

**Compliance With Llm Reviewing Policy:**

Affirmed.

**Final Justification:**

I have carefully reviewed the authors' additional response and appreciate their clarifications. Most of my concerns have been addressed. However, I believe that several of these points would require substantial revisions to the current manuscript.

Therefore, I will update my score from Reject to Weak Reject. While I do not feel strongly opposed to acceptance, I believe that if the paper is ultimately accepted, it is important that all the points discussed during the rebuttal are properly incorporated in to the final manuscript.

**Key Questions For Authors:**

1. [Weaknesses 1] Please clarify the relationship between the DDPO per-step log-likelihood in Theorem 2.1 and the DDPO objective in Eq. (5), particularly how the advantage term is treated in the analysis.

2. [Weaknesses 2] Please clarify the methodological novelty and naming of AWM, especially in relation to DDPO, which already incorporates advantage-weighted updates.

3. [Weaknesses 3] Please clarify the experimental inconsistencies noted in the figures and ablation results, and whether variance across runs should be reported for the main tables.

4. [Weaknesses 4] Please provide experimental evidence directly supporting the variance analysis in Theorem 2.3, beyond the current evaluation based on final image metrics.

**Limitations:**

The paper does not explicitly discuss limitations of the proposed method. In particular, although the method is motivated by a theoretical variance analysis, the connection between the simplified theoretical setting and large-scale RL fine-tuning of modern text-to-image diffusion models is not fully established.

**Strengths And Weaknesses:**

### [Strengths]

1. The problem addressed in this paper, reinforcement learning-based fine-tuning for diffusion models, is an actively studied and important research direction. Their attempt to improve training efficiency by connecting RL fine-tuning with theoretical analysis is timely and has the potential to contribute meaningfully to this area.

2. The theoretical observation that matching using noisy states can lead to higher variance compared to matching using clean and forward-diffused states, while remaining unbiased, is an insightful point. This analysis is not only relevant to the proposed method but may also be useful for understanding and improving other diffusion model research.

3. The experimental results are compelling. The paper reports that the proposed method achieves better performance while requiring significantly fewer training hours than existing approaches. Such improvements in training efficiency are practically important for large-scale diffusion models.

### [Weaknesses]

1. Unclear relationship between Theorem 2.1 and the actual DDPO objective in Eq. (5).
  * Theorem 2.1 states an equivalence between DDPO's per-step likelihood objective and a denoising score matching objective. However, in the actual DDPO formulation in Eq. (5), the optimization is weighted by the reward term $r-b$, i.e., the advantage, which depends on $x_0$. Because this term is sample-dependent, it does not seem valid to simply ignore it even at the expectation level. For this reason, the paper should clarify much more explicitly what is meant by the "DDPO per-step log-likelihood" in Theorem 2.1, how it relates to the full RL objective in Eq. (5), and whether the proposed denoising score matching interpretation still holds once the advantage term is included.

2. The naming and conceptual distinction of "Advantage Weighted Matching" are somewhat unclear.
  * As I understand it, DDPO already incorporates advantage-weighted updates through the reward term in Eq. (5). From this perspective, the main difference of the proposed method seems not to be the use of advantage weighting itself, but rather the fact that it performs matching on $(x_0,x_t)$ instead of $(x_t, x_s)$. This is also consistent with the paper's own discussion of the key differences on page 6, which emphasizes variance reduction and the decoupling of training from sampling. Since "reward-derived weights" already appear to be present in DDPO as well, the paper would benefit from a clearer explanation of what the true methodological novelty is and why the proposed name is the most appropriate one.

3. Several experimental results raise questions that are not sufficiently discussed.
  * In Figures 5(a) and 5(b), AWM appears to start from a worse initial performance than SD3.5M, which is deserves explanation.
  * Figures 7 and 8 present ablation studies, but the configurations selected based on these ablations still appear to underperform compared to the main table results. Therefore, it is unclear what accounts for this discrepancy. If the difference is due to run-to-run variation or different experimental settings, then the main results should ideally include error bars with repeated runs.

4. The variance claim in Theorem 2.3 is not directly validated experimentally.
  * Theorem 2.3 is one of the central theoretical claims of the paper, as it argues that the noisy-target formulation has higher variance. However, the supporting experiments evaluate this claim only indirectly through final image-generation quality, such as FID. It would strengthen the paper substantially if they provided experiments that directly target the claimed phenomenon, for example by measuring variance or some closely related statistic during training.

5. Eq. (13) needs additional explanation.
  * In Eq. (13), the gradient of the log-likelihood for a specific sample $x_i$ is written in the form of an expectation (second term and the last term). It is unclear what this expectation is taken over, and how exactly this expression is derived. A clearer derivation and explicit notation for the expectation would improve readability and avoid confusion.

6. The presentation and notation are not sufficiently consistent.
* The notation changes across sections and some symbols appear before they are properly defined. I list some examples below.
  * In Section 2 the policy is written as $p$, whereas in Section 3 it is written as $\pi$.
  * In Eq. (5), terms such as $p_{\theta_{old}}$, $c$, and $r$ appear before they are clearly defined, with some of these definitions only introduced later in Section 3.
  * In Theorem 2.3, a covariance inequality is stated, but the meaning of the matrix inequality should be explicitly defined.
  * The DDPO formulation mixes a baseline-based presentation with GRPO-style advantage computation, which makes the exposition less consistent.
  * On page 5, there is a table in the middle of the page without a caption and table number.
  * The notation $x_i$ is overloaded, as subscripts are used both for diffusion time indices and for sample indices.
  * "Figure 7(a-c) in line 429 should likely be "Figures 7 and 8."
  * Eqs. (15) and (16) appear to contain unnecessary commass.

---

> ### Author Rebuttal · Authors · 2026-03-31
>
> **Weakness 1:** What is the relationship between Theorem 2.1 and Eq. (5), which includes the advantage term?
>
> **Answer 1:** We thank the reviewer for raising this important point. We agree that Theorem 2.1 directly analyzes the inner per-step DDPO likelihood term, not the full advantage-weighted objective in Eq. (5). We next show that the Section 2 analysis extends to the advantage-weighted setting.
>
> Fix one update and one timestep $t$, and write $s=t-\Delta t$. Let the frozen rollout distribution be
> $$
> q_{\bar\theta}(c,x_0,x_s,x_t)
> :=
> p(c)\pi_{\bar\theta}(x_0\mid c)p(x_s\mid x_0)p(x_t\mid x_s),
> \qquad \bar\theta=\theta_{\mathrm{old}},
> $$
> and let $A(x_0,c)$ denote the stop-gradient advantage.
>
> The generalized equivalence can then be stated in two steps.
>
> First, Theorem 2.1 can be generalized for any outer distribution of $x_0, c$. This is because the theorem only depends on the induced joint distribution of $(x_s,x_t)$. Second, Lemma 2.2 uses $\int p(x_t|x_s)p(x_s)dx_s=p(x_t)$, which remains valid after conditioning on $x_0$. Hence, both steps are still valid after weighting by stop-gradient $A(x_0,c)$ and averaging over the frozen rollout distribution.
>
> Combining the two steps, we obtain
> $$\nabla_\theta L_t^{\mathrm{DDPO}}(\theta;\bar\theta)=a_t\nabla_\theta L_t^{\mathrm{AWM}}(\theta;\bar\theta).$$
>
> The core proof idea is to condition on $x_0$ and use the law of total expectation, which allows the stop-gradient advantage term $A(x_0,c)$ to be separated from the inner noisy-to-clean equivalence. **We omit the full derivation here due to char limit and can provide it if helpful.**
>
> **Weakness 2:** Why call the method “Advantage Weighted Matching” if DDPO also uses advantage-weighted updates?
>
> **Answer 2:** We agree that using advantages in policy gradients is standard and not our contribution. The novelty lies in **what is being weighted**: DDPO weights a step-wise noisy-target reverse likelihood, whereas AWM weights the pretraining-compatible clean-target score/flow-matching objective. That is why “Matching,” rather than “Advantage,” is the key part of our method name.
>
>
> **Weakness 3:** The figures and ablations appear inconsistent.
> **Answer 3:**
>
> >AWM appears to start from a worse initial performance than SD3.5M
>
> This is because we use cfg=1.0 during sampling as stated in section 'Experiment Settings' in Appendix F, while SD3.5M baseline use cfg=4.5. We will make this setting clear in the main paper.
>
> >Ablations still appear to underperform compared to the main table results.
>
> Thank you for pointing this out. The earlier ablations were run with smaller-scale / earlier configurations, whereas the main tables use the final stronger setting. We provide three additional ablations under the main-result configuration. Note that the first row of each ablation tables below is consistent with our main results (Figure 1 & 5, and Table 1 in the main paper).
>
> |GenEval: step|50|100|200|300|400|500|600|
> |-|-|-|-|-|-|-|-|
> |AWM with $T=6$ (final)|0.890|0.927|0.957|0.970|0.975|0.978|0.980|
> |AWM with $T=4$ |0.860|0.901|0.928|0.950|0.961|0.960|0.972|
>
> |GenEval: step|50|100|200|300|400|500|600|
> |-|-|-|-|-|-|-|-|
> |AWM with $\theta_{\text{old}} = \theta$ (final)|0.890|0.927|0.957|0.970|0.975|0.978|0.980|
> |AWM with $\theta_{\text{old}} = \theta_{\text{EMA}}$ + Importance Sampling|0.886|0.947|0.963|0.970|0.974|0.976|0.975|
>
> |GenEval: step|50|100|200|300|400|500|600|
> |-|-|-|-|-|-|-|-|
> |AWM with $\beta=1$ (final)|0.890|0.927|0.957|0.970|0.975|0.978|0.980|
> |AWM with $\beta=3$|0.880|0.931|0.960|0.954|0.956|0.967|0.967|
>
>
> **Weakness 4:** The variance claim in Theorem 2.3 is not directly validated.
> **Answer 4:** We agree that this point should be validated more directly. We therefore directly quantify the conditional target variance for the clean-target and noisy-target (DDPO) formulations on CIFAR-10 and ImageNet-64. Our measurements across 10 log-spaced noise levels confirm this consistently on both datasets:
>
> | |σ=0.002|σ=0.006|σ=0.019|σ=0.057|σ=0.173|σ=0.529|σ=1.61|σ=4.92|σ=15.0|σ=80.0|
> |-|-|-|-|-|-|-|-|-|-|-|
> |CIFAR-10 clean|0.00e+00 | 0.00e+00 | 0.00e+00 | 0.00e+00 | 0.00e+00 | 0.00e+00 | 4.12e-06 | 126 | 597 | 694 |
> |CIFAR-10 noisy|0.02 | 0.15 | 1.37 | 12.74 | 119 | 1104 | 10274 | 95759 |890747|2.53e+07|
> |ImageNet-64 clean|0.00e+00|0.00e+00|0.00e+00|0.00e+00|0.00e+00|0.00e+00|0.00e+00|0.03|1181|2665|
> |ImageNet-64 noisy|0.06|0.59|5.48|50.96|474|4415|41097|382533|3.56e+06|1.01e+08|
>
> **Weakness 5 & Weakness 6:** Eq. (13) and the notation need clarification.
> **Answer 5:** We thank the reviewer for carefully identifying these presentation and notation issues. We will revise the paper to clarify that Eq. (13) takes expectation over the forward-noising randomness $(t,\epsilon)$, unify the policy notation ($\pi$ and $p$), define terms in Eq. (5) earlier, explicitly state that the covariance inequality in Theorem 2.3 is in the positive semidefinite (Loewner) order, and fix the remaining formatting and cross-reference issues.

---

> > ### Author Rebuttal · Reviewer_i7aJ · 2026-04-04
> >
> > Thank you for the authors’ response. The additional experimental results are helpful in addressing my concerns in W3 and W4. However, my main concerns are still not fully resolved.
> >
> > **W1**. The authors argue that Theorem 2.1 only depends on $(x_s,x_t)$, but the additional weighting term $r(x_0,c)$ depends on $x_0$, which is generated through the sequential sampling of $x_t$. Therefore, $r(x_0,c)$ is ultimately dependent on $x_t$. From this perspective, the connection is not straightforward, since the theorem does not explicitly account for this dependency on $x_0$. It would be helpful if the authors could provide a more formal derivation that incorporates this point and clearly shows how Eq. (5) follows under this dependency.
> >
> > **W2**. Personally, I am somewhat concerned whether the distinction based on the clean-target score/flow-matching objective is best captured by the term "matching." That said, the naming itself is not a critical issue. It would be more important for the paper to more clearly highlight and articulate the claimed distinction.

---

> > > ### Author Response · Authors · 2026-04-04
> > >
> > > **Weakness 1:** Rigorous full proof
> > >
> > > **Answer 1:**
> > > Let $k=0,\dots,T$ index reverse-time states, and consider the on-policy $\bar\theta := \theta_{\mathrm{old}} = \text{stopgrad}(\theta)$ setting. Fix one DDPO step, and write $s=t-\Delta t$.
> > > Define the reverse-rollout law
> > > $$Q ^ {\mathrm{rev}} _ {\bar{\theta}}(c,x_{0:T})
> > > := p(c)p(x_T)\prod_{k=1}^T p_{\bar\theta}(x_{k-1}\mid x_k,c).$$
> > > Define the terminal advantage
> > > $$A(x_0,c):=r(x_0,c)-b(c).$$
> > > Crucially, $A(x_0,c)$ is not assumed independent of $x_t$; it is simply a measurable function of $(x_0,c)$.
> > >
> > > Now isolate one step of Eq. (5):
> > > $$J_t(\theta;\bar\theta):=\mathbb{E} _ { Q ^ { \mathrm{rev} } _ { \bar{\theta} } }\left[A(x_0,c)\frac{p_\theta(x_s\mid x_t,c)}{p_{\bar\theta}(x_s\mid x_t,c)}\right].$$
> > > Then the local gradient at the frozen policy is
> > > $$\nabla _ \theta J _ t(\theta;\bar{\theta})\big|_{\theta=\bar{\theta}}=\mathbb E _ {Q ^ {\mathrm{rev}} _ {\bar{\theta}}}\left[A(x_0,c)\nabla _ \theta \log p _ \theta(x_s\mid x_t,c)\big| _ {\theta=\bar\theta}\right].\tag{1}$$
> > > This is the REINFORCE-style statement: it comes from
> > >
> > > $$\rho_t^\theta:=\frac{p_\theta}{p_{\bar\theta}}, \quad \nabla_\theta \rho_t^\theta=\rho_t^\theta \nabla_\theta \log p_\theta,\qquad\rho_t^{\bar\theta}=1,$$
> > >
> > > The gradient is unchanged either analyzing $r(x_0,c)$ or $A(x_0,c)=r(x_0,c)-b(c)$ because
> > > $$\mathbb E _ {c,x_t} \left[ b(c)\int \nabla_\theta p_\theta(x_s\mid x_t,c)\big| _ {\theta=\bar{\theta}}dx_s\right]=\mathbb E_{c,x_t}\left[b(c)\nabla_\theta \int p_\theta(x_s\mid x_t,c)dx_s\big|_{\theta=\bar\theta}\right]=0.$$
> > > Now define the marginal of $Q^{\mathrm{rev}} _ {\bar{\theta}}(c,x _ {0:T})$
> > >
> > > $$Q ^ {(0,s,t)} _ {\bar\theta} := \mathrm{Law} _ {Q ^ {\mathrm{rev}} _ {\bar\theta}}(c,x_0,x_s,x_t).$$
> > >
> > > This marginal can equivalently be written in forward form due to Lemma C.1 in paper
> > >
> > > $$Q ^ {(0,s,t)} _ {\bar\theta}(c,x_0,x_s,x_t)=p(c)\pi _ {\bar\theta}(x_0\mid c)p(x_s\mid x_0)p(x_t\mid x_s),\tag{2}$$
> > >
> > > where $\pi_{\bar\theta}(x_0\mid c)$ is the terminal distribution induced by the joint. This is a factorization of the same joint law; it does not assert independence between $x_0$ and $x_t$.
> > >
> > > Using this marginal, (1) becomes
> > >
> > > $$\mathbb E _ {Q^{(0,s,t)} _ {\bar\theta}}\left[A(x_0,c)\nabla_\theta \log p_\theta(x_s\mid x_t,c)\big|_{\theta=\bar\theta}
> > > \right].\tag{3}$$
> > >
> > > Now condition on $(x_0,c)$, it equals
> > > $$\mathbb E_{c,x_0}\left[A(x_0,c)\mathbb E_{x_s,x_t\mid x_0,c}\left[\nabla_\theta \log p_\theta(x_s\mid x_t,c)\big|_{\theta=\bar\theta}\right]\right].\tag{4}$$
> > >
> > > We do not say $A(x_0,c)$ is independent of $x_t$. We keep the joint law intact, then condition on $(x_0,c)$. The derivation in Appendix C.2 shows that for each $(x_s,x_t)|(x_0,c)$,
> > >
> > > $$\log p_\theta(x_s\mid x_t,c)=-\alpha_t\ell^{\mathrm{noisy}}_{\theta,t}(x_s,x_t,c)+\kappa_t,\qquad \alpha_t>0,\tag{5}$$
> > >
> > > where $\kappa_t$ is $\theta$-independent, $\nabla_\theta \kappa_t=0$, and
> > >
> > > $$\ell^{\mathrm{noisy}} _ {\theta,t}(x_s,x_t,c):=\bigl \lVert s_\theta(x_t,t,c)-\nabla _ {x_t}\log p(x_t\mid x_s)\bigr \rVert ^2.$$
> > >
> > > Therefore
> > > $$\mathbb E_{x_s,x_t\mid x_0,c}\left[\nabla_\theta \log p_\theta(x_s\mid x_t,c)\right]=-\alpha_t\nabla_\theta
> > > \mathbb E_{x_s,x_t\mid x_0,c}\left[\ell^{\mathrm{noisy}}_{\theta,t}(x_s,x_t,c)\right].\tag{6}$$
> > >
> > > Now define the clean-target conditional objective
> > >
> > > $$\ell^{\mathrm{clean}} _ {\theta,t}(x_0,x_t,c):=\bigl\lVert s _ \theta(x_t,t,c)-\nabla _ {x_t}\log p(x_t\mid x_0)\bigr\rVert^2.$$
> > >
> > > Then the conditional version of Lemma 2.2 is:
> > >
> > > $$\nabla _ \theta\mathbb E _ {x_s\sim p(\cdot\mid x_0)}\mathbb E _ {x_t\sim p(\cdot\mid x_s)}\left[\ell^{\mathrm{noisy}} _ {\theta,t}(x_s,x_t,c)\right]=\nabla _ \theta\mathbb{E} _ {x_t\sim p(\cdot \mid x_0)}\left[\ell^{\mathrm{clean}} _ {\theta,t}(x_0,x_t,c)\right].\tag{7}$$
> > >
> > > The proof is exactly the Appendix-C.1 argument, but with
> > > $$p(x_s)\to p(x_s\mid x_0),\qquad p(x_t)\to p(x_t\mid x_0),$$
> > > using
> > > $$\int p(x_t\mid x_s)p(x_s\mid x_0)dx_s=p(x_t\mid x_0).\tag{8}$$
> > >
> > > Plugging (6) and (7) into (4) yields
> > >
> > > $$\nabla _ \theta J_t(\theta;\bar\theta)\big| _ {\theta=\bar\theta}=-\alpha_t \nabla_\theta
> > > \mathbb E _ {p(c)\pi_{\bar\theta}(x_0\mid c)p(x_t\mid x_0)}\left[A(x_0,c)\ell^{\mathrm{clean}} _ {\theta,t}(x_0,x_t,c)\right] _ {\theta=\bar\theta}.\tag{9}$$
> > >
> > > Summing adjacent steps over $t$ gives the full local-gradient relation for Eq. (5) in paper:
> > > $$\boxed{\nabla _ \theta J _ {\mathrm{DDPO}}(\theta;\bar\theta)\big| _ {\theta=\bar\theta}=-\sum_t \alpha_t
> > > \nabla _ \theta\mathbb E _ {p(c)\pi _ {\bar\theta}(x_0\mid c)p(x_t\mid x_0)}\left[A(x_0,c)\ell ^ {\mathrm{clean}} _ {\theta,t}(x_0,x_t,c)\right] _ {\theta=\bar\theta}.}$$
> > > This is the theoretical clean-target advantage-weighted DSM that motivates AWM.
> > >
> > > **Weakness 2:** clearly highlight and articulate the claimed distinction.
> > >
> > > **Answer 2:** Thank you for the clarification. We agree that name is not the main point. We agree this point should be stated more clearly, and we will revise the abstract/introduction/method overview to state this earlier and more explicitly.

---

### Official Review · Reviewer_t8kk · 2026-03-12

**Soundness:** 3
**Presentation:** 4
**Significance:** 2
**Originality:** 3
**Overall Recommendation:** 4
**Confidence:** 4

**Summary:**

In this paper, the authors introduce a theoretical work to align the gap in using RL post-training for diffusion models. The motivation is that the existing post-training methods like DDPO do not align their training objectives with those of the diffusion model pre-training (which is a standard practice for LLM-based post-training). The existing methods use a multi-step MDP using noisy targets during both training and sampling, which increases gradient variance and slows the convergence. Based on this intuition, the authors formulate a theoretical equivalence that DDPO is performing denoising score matching with noisy targets, and thus, a one-step jump could be applied directly to a clean data target during the RL weight update. By reweighting the pretraining loss with advantages, the proposed work unifies the RL post-training with its diffusion pretraining objective. Empirically, the authors demonstrate an up to a 34 times speedup built upon SD3.5M and FLUX.

**Compliance With Llm Reviewing Policy:**

Affirmed.

**Final Justification:**

The authors have addressed my concerns in our rebuttal discussion. As a result, I am happy to raise my score.

Please include all the results in the author response in the camera-ready version if it is accepted.

**Key Questions For Authors:**

1. Since the proposed derivation of AWM is a reformulation of the multi-step MDP of DDPO into a sequence-level conditional policy that jumps in one step to $x_0$. Considering, the proposed reformulation is highly related to both reward weighted regression and the one-step diffusion distillation, could you clarify why the proposed work is not just a simple reformulation of one-step diffusion distillation using reward weighted regression?

2. Does the discretization error in DDPO contribute to the variance more than the noisy target itself?

3. How do you justify the forward-reverse Gaussian identity assumption when the samples used for the gradient update come from a non-Gaussian or non-stationary policy distribution?

4. While the authors demonstrate the sample diversity after 600 steps, the proposed reformulation makes the RL post-training a one-step distillation. However, the authors still use the multi-step solver during the sampling, which leads to a clear mismatch. Could you conduct ablation studies to show the performance if a one-step sampler is used? Meanwhile, have you performed long-term training comparisons (beyond 1000 steps) to see if the AWM's efficiency comes at the cost of a narrower distribution compared to the DDPO?

**Limitations:**

yes

**Strengths And Weaknesses:**

**Strengths:**

1. The empirical results are promising. The authors show significant wall-clock savings across multiple benchmarks (e.g., the same GenEval quality 16.54 times faster than Flow-GRPO).

2. The authors provide in-detailed theoretical analysis to align the RL post-training of diffusion models with that of the LLM, which reduces the need for multi-step noise sampling during RL post-training.

3. By replacing the noisy intermediate targets with the clean data, the authors provide a rigorous proof (Theorem 2.3) that the conditional covariance of the estimator is reduced, which leads to more efficient and sample optimization.

**Weaknesses:**

1. Based on my understanding, while the authors present a complex alignment theory, the core behind their theoretical work is a one-step jump update, which has been extensively studied in the one-step distillation of diffusion models. Meanwhile, the authors borrow the well-studied concepts from the reward weighted regression and reformulate them under the “AWM” name without acknowledging this leap. However, the authors did not include any related work on the one-step distillation of diffusion models and the reward weighted regression. The authors should include a detailed related work section to explicitly discuss how their formulation differs from the existing one-step distillation and reward weighted regression methods.

2. The proof of equivalence (Theorem 2.1) assumes an ideal setting where the discretization error can be ignored. However, this does not hold in practice. Meanwhile, as far as I know, this is also the main challenge for the one-step distillation for diffusion models. The authors should conduct further theoretical analysis to prove that such discretization (or truncation) error will not make the proposed work invalid.

3. The proposed theoretical work builds upon the assumption that the forward Gaussian process is identical to that of the reverse one. However, in control-based RL (main tasks are trajectory planning), the samples generated by the reverse policy are constantly shifting, which might violate the static distribution assumptions used in the proofs. The authors should explicitly explain why this will not be a problem in image generation tasks even though they build upon the RL.

4. The proposed formulation treats DDPO as a one-step distillation to justify the switch to their AWM. However, it still requires multi-step sampling to generate the clean images for reward evaluation. It does not make sense to me to only simplify the gradient update but not the sampling cost. Could the authors conduct ablation experiments to show the generation quality when a one-step sampler is used?

---

> ### Author Rebuttal · Authors · 2026-03-31
>
> ## Relationship with One-step Distillation
>
> We would like to first clarify a central point that may have caused confusion. AWM does **not** use a distilled one-step model. In our paper, the clean sample $x_0$ is always generated by standard **multi-step** sampling with ODE/SDE solvers (e.g., Euler, Euler-Maruyama).
>
> Likewise, when AWM is described in the Problem Setup Comparison table (in Page 5) as optimizing the sample probability $p_\theta(x_0 \mid c)$, this is estimated through a standard **multi-step** flow-matching model $v_\theta$, as made explicit in Eq. (10). In other words, AWM should be understood as an **RL training method** while keeping the underlying model and inference procedure **multi-step**, not as a form of one-step distillation.
>
> We do think that this distinction matters because many of the introduced questions are due to this one-step assumption, **including Weaknesses 1, 2, 4 and Questions 1, 4** as mentioned below.
>
> ## Relationship with Reward Weighted Regression
>
> We respectfully disagree that AWM is merely a direct application of standard Reward-Weighted Regression (RWR). While the naming is similar, AWM is conceptually and technically different from RWR and AWR.
>
> Classical RWR[1] is derived from a KL-constrained policy improvement objective around a behavior policy $\mu$:
>
> $$\max_{\pi} E_{s \sim d_{\mu}, a \sim \pi(.|s)}[r(s,a)]- \beta E_{s \sim d_{\mu}}[D_{KL}(\pi(.|s) \| \mu(.|s))].$$
>
> Its optimal policy is
> $$\pi^*(a|s)= \frac{1}{Z(s)}\mu(a|s)\exp\left(\frac{r(s,a)}{\beta}\right),$$
>
> which leads to the regression form
>
> $$\pi^*(a|s) = \arg\max_{\pi} E_{s \sim d_\mu, a \sim \mu(.|s)} [ \log \pi(a|s) \exp(r(s,a)/\beta) ].$$
>
> This is exactly Reward-Weighted Regression. Replacing $r(s,a)$ by the advantage $A^\mu(s,a)$ yields Advantage-Weighted Regression:
> $$\pi^* = \arg\max_{\pi} E_{s \sim d_\mu, a \sim \mu(.|s)}[ \log \pi(a|s) \exp(A^\mu(s,a)/\beta) ].$$
>
>
> Therefore, both RWR and AWR rely on **exponential reweighting with strictly positive weights**. In contrast, AWM is derived from a policy-gradient objective and directly uses the **signed** advantage $A$ to weight the pretraining score/flow-matching objective. This linear weighting is consistent with the standard Policy Gradient objective $\mathbb{E}[\nabla \log \pi \cdot A]$, making AWM mathematically distinct from the density-matching RWR approaches.
>
> [1] Relative Entropy Policy Search.
>
> [2] Advantage-Weighted Regression: Simple and Scalable Off-Policy Reinforcement Learning.
>
> **Weakness 1 & Question 1:** Is AWM mainly a reformulation of reward-weighted regression or one-step distillation?
>
> **Answer 1:** In short, AWM is neither a one-step distillation method nor a standard RWR/AWR objective: it keeps multi-step inference and changes the matching objective being weighted. We provide the two clarifications above for detail.
>
> **Weakness 2:** Theorem 2.1 ignores discretization error.
>
> **Answer 2:** Thank you for pointing this out. Analyzing the idealized setting without discretization error is a deliberate and standard theoretical choice: it establishes a **best-case baseline** for DDPO. In other words, this is not a simplification in our analysis, but rather underscores a **core, inherent flaw of the DDPO algorithm itself**, since DDPO relies on a solver-discretized reverse transition $p_\theta(x_{t-\Delta t}\mid x_t)$. Our theorem analyze an idealized/oracle DDPO without discretization error; even there, the noisy-target issue remains, while discretization bias is an additional solver-dependent limitation of DDPO. AWM, in contrast, is defined on forward-process pairs $(x_0,x_t)$ and avoids reverse-solver discretization.
>
> **Weakness3 & Question 3:** Why is the forward-reverse identity still valid under a changing RL policy?
>
> **Answer 3:** Thank you for the helpful comment. Our analysis is a **per-iteration** statement: the forward/reverse equivalence is applied for each fixed current policy $\theta$ separately, rather than assuming a single static distribution over the whole RL process. Therefore, although the policy and induced sample distribution change across iterations, this does not invalidate the proof. In image generation, the forward noising kernel itself remains fixed; only the model-induced data distribution evolves during training.
>
> **Weakness4 & Question 4:** Why not evaluate a one-step sampler or much longer training?
>
> **Answer 4:** Please see Relationship with One-step Distillation.
>
> **Question 2:** Does the discretization error in DDPO contribute to the variance more than the noisy target itself?
>
> **Answer 5:** Thank you for this question. We would like to distinguish these two effects. In our analysis, discretization error in DDPO is primarily a **bias** term, since it comes from approximating the reverse-time SDE. By contrast, Theorem 2.3 studies the variance effect of the noisy target itself through its larger conditional covariance. We therefore do not claim which one is larger quantitatively.

---

> > ### Author Rebuttal · Reviewer_t8kk · 2026-04-03
> >
> > 1. Although the authors clarified that AWM is an RL method, they did not provide results for the suggested ablation (using a one-step sampler). Since AWM decouples training from sampling, I would like to see the ablation of training the model with a 20-step sampler but evaluated with a 1-step ODE.
> >
> > 2. Could the authors provide a quantitative comparison of the variance introduced by a noisy target v.s. the bias introduced by discretization error in a standard 20-step DDPO?
> >
> > 3. What is the precise ratio of time spent on the rollout v.s. the gradient update phase in your reported wall-clock experiments?
> >
> > 4. In the new ablations provided for other reviewers, AWM at step 50 starts at a GenEval score of 0.89. Could the authors explain why this starting point is significantly higher than the base SD3.5M model score (0.63) reported in Table 1?

---

> > > ### Author Response · Authors · 2026-04-03
> > >
> > > **Question 1:** Ablation of training the model with a 20-step sampler but evaluated with a 1-step ODE.
> > >
> > > **Answer 1:** The model fine-tuned by AWM remains a standard multi-step diffusion/flow model and does not acquire one-step generation capability. **We visualized 1-step ODE samples from the AWM-trained model and observed blurry outputs, consistent with the behavior of evaluating diffusion/flow model with 1-step ODE**. One-step generation would require dedicated step distillation, which is orthogonal to AWM.
> > >
> > > To make sure we fully address your concern, could we kindly ask for clarification on the motivation behind this question? One possible reason we can think of is that in Figure 4 we state AWM supports "any sampler". To clarify: by "any sampler" we refer to multi-step diffusion/flow samplers such as DDPM, DDIM, DPM-Solver, and UniPC. These are all multi-step solvers that require tens of function evaluations. AWM decouples the training objective from the choice of sampler, but it does not change the fundamental nature of the model as a multi-step diffusion model.
> > >
> > > **Question 2:** a quantitative comparison of the variance introduced by a noisy target v.s. the bias introduced by discretization error in 20-step DDPO?
> > >
> > > **Answer 2:**
> > >
> > > We thank the reviewer for this insightful question. We provide a quantitative comparison below.
> > >
> > > Table 1: Target Variance
> > >
> > > | | σ=0.002 | σ=0.006 | σ=0.019 | σ=0.057 | σ=0.173 | σ=0.529 | σ=1.61 | σ=4.92 | σ=15.0 | σ=80.0 |
> > > |-|-|-|-|-|-|-|-|-|-|-|
> > > | **CIFAR-10 clean** | 0.00e+00 | 0.00e+00 | 0.00e+00 | 0.00e+00 | 0.00e+00 | 0.00e+00 | 4.12e-06 | 126 | 597 | 694 |
> > > | **CIFAR-10 noisy** | 0.02 | 0.15 | 1.37 | 12.74 | 119 | 1104 | 10274 | 95759 | 890747 | 2.53e+07 |
> > > | **ImageNet-64 clean** | 0.00e+00 | 0.00e+00 | 0.00e+00 | 0.00e+00 | 0.00e+00 | 0.00e+00 | 0.00e+00 | 0.03 | 1181 | 2665 |
> > > | **ImageNet-64 noisy** | 0.06 | 0.59 | 5.48 | 50.96 | 474 | 4415 | 41097 | 382533 | 3.56e+06 | 1.01e+08 |
> > >
> > > **Discretization error (Table 2).** We measure the discretization error of the 20-step DDPM sampler used in DDPO. Specifically, we use a 381-step sampler as the reference (where the 20-step grid is an exact nested subset, since $381 = 19 \times 20 + 1$). For each of the 20 coarse steps at noise level $\sigma_k$, we start from the 381-step reference trajectory at $\sigma_k$, take one coarse step with exactly coupled Brownian noise, and report $\mathbb{E}[\|x_{\text{20-step}} - x_{\text{381-step}}\|^2]$ averaged over 64 trajectories.
> > >
> > > Table 2a: 20-Step bias — CIFAR-10 (vs 381-step reference)
> > >
> > > | | σ=80.0 | σ=59.7 | σ=43.9 | σ=31.9 | σ=22.8 | σ=16.0 | σ=11.1 | σ=7.47 | σ=4.93 | σ=3.17 |
> > > |-|-|-|-|-|-|-|-|-|-|-|
> > > | **bias** | 0.56 | 0.89 | 1.60 | 2.86 | 6.80 | 7.02 | 8.30 | 10.85 | 12.28 | 13.90 |
> > >
> > > | | σ=1.98 | σ=1.19 | σ=0.693 | σ=0.384 | σ=0.201 | σ=0.099 | σ=0.045 | σ=0.018 | σ=0.007 | σ=0.002 |
> > > |-|-|-|-|-|-|-|-|-|-|-|
> > > |**bias**|11.24|8.75|5.60|3.49|2.02|1.04|0.46|0.15|0.03|0.00|
> > >
> > > Table 2b: 20-Step bias — ImageNet-64 (vs 381-step reference)
> > >
> > > | | σ=80.0 | σ=59.7 | σ=43.9 | σ=31.9 | σ=22.8 | σ=16.0 | σ=11.1 | σ=7.47 | σ=4.93 | σ=3.17 |
> > > |-|-|-|-|-|-|-|-|-|-|-|
> > > | **bias** | 6.35 | 12.45 | 18.48 | 19.90 | 28.09 | 39.46 | 51.13 | 41.47 | 40.33 | 38.85 |
> > >
> > > | | σ=1.98 | σ=1.19 | σ=0.693 | σ=0.384 | σ=0.201 | σ=0.099 | σ=0.045 | σ=0.018 | σ=0.007 | σ=0.002 |
> > > |-|-|-|-|-|-|-|-|-|-|-|
> > > | **bias** |32.00|25.67| 19.13 | 12.75 | 7.54 | 3.85 | 1.63 | 0.55 | 0.14 | 0.00 |
> > >
> > > **Comparison.** Across all noise levels and both datasets, the noisy-target variance exceeds the 20-step discretization bias.
> > >
> > > **Question 3:** the precise ratio of time spent on the rollout v.s. the gradient update phase
> > >
> > > **Answer 3:** In our experiments, the rollout (forward pass) uses 14 steps without CFG, and the gradient update (backward pass) uses 6 timesteps. Since the backward pass requires roughly 2x the compute per step compared to the forward pass, the ratio between rollout and gradient update is approximately **14:12** (~53.8% : 46.2%). In practice, we measured ~21s for rollout and ~17.5s for gradient update per iteration, giving an empirical ratio of ~54.5% : 45.5%, consistent with this estimate.
> > >
> > >
> > > **Question 4:** AWM at step 50 starts at a GenEval score 0.89. Why this starting point is higher than the base SD3.5M model score (0.63)?
> > >
> > > **Answer 4:** The GenEval score of 0.63 in Table 1 is evaluated with CFG scale 4.5, which is the default setting for SD3.5M. However, in our RL training we do not use CFG during sampling (i.e., CFG=1, pure conditional sampling). Under this no-CFG setting, the base SD3.5M GenEval score is **0.24**, which is the true starting point of our RL training. The score of 0.89 at step 50 reflects rapid improvement during the early phase of training. We provide the full training progression below:
> > >
> > > | Step | 0 | 10 | 20 | 30 | 40 | 50 |
> > > |-|-|-|-|-|-|-|
> > > | **GenEval** | 0.24 | 0.69 | 0.78 | 0.85 | 0.87 | 0.89 |
> > >
> > > As shown, AWM improves GenEval from 0.24 to 0.69 within just 10 steps, demonstrating fast early convergence.

---

### Official Review · Reviewer_a4zM · 2026-03-23

**Soundness:** 3
**Presentation:** 2
**Significance:** 3
**Originality:** 3
**Overall Recommendation:** 4
**Confidence:** 2

**Summary:**

The paper frames DDPO as a score/flow matching scheme and shows that it has high variance due to its noisy stepwise targets. The authors suggest an alternative approach that does not have a sequential formulation nor utilises noisy targets, leading to compute savings and to lower variance under some mild assumptions.

**Compliance With Llm Reviewing Policy:**

Affirmed.

**Final Justification:**

I thank the authors for their interaction and the clarifications they made, reinforcing my initial assessment. I agree with the other reviewers that a certain effort should be put in the manuscript to incorporate the many clarifications while enhancing readability.

**Key Questions For Authors:**

How does the approach fare in efficiency to non-flow based methods?

**Limitations:**

yes

**Strengths And Weaknesses:**

The paper bases its results on theoretical analysis of the state-of-the-art, linking DDPO to score/flow matching and showing its high variance when seen from this perspective. The analysis, though, simplifies the basic algorithm by ignoring error terms related to discretisation and bias. The paper then suggests an alternative approach that has lower variance by relating to the actual generated target rather than the noised data points. The ablation study section does not seem to add value to the paper as it does not critically target the approach's design choices.

The paper is well presented but could be improved. Some acronyms are not expanded when they appear first. The detailed description of DDPO in the appendix does not add much detail to the section in the paper itself. Figure 2 is not referenced in the body of the paper.

The empirical results show significant improvement in performance for the proposed method in GenEval, OCR and PickScore tasks. Also, it shows  improved  efficiency when compared to an alternative flow matching baseline (Flow-GRPO, TempFlow-GRPO). However, this is not shown for other baselines that are not particularly flow-matching based (e.g. DDPO-based frameworks).

The paper presents and interesting theoretical angle, and the approach appears to beat other flow-matching RL baselines in performance and efficiency.

---

> ### Author Rebuttal · Authors · 2026-03-31
>
> **Question 1:** How does AWM compare in efficiency to non-flow-based methods?
> **Answer 1:**
>
>
> We additionally reproduce DDPO on SD3.5 Medium and compare the compute required to reach matched quality thresholds. The results are shown below.
>
> |  | AWM GPU hours | DDPO GPU hours| speedup |
> |---|---|--|--|
> | GenEval=0.90|54.44| 1568.24 | 28.81$\times$ |
> | PickScore=23.0 | 26.63 | 945.31 | 35.50$\times$ |
> | OCR=0.85 | 18.30| 387.74  | 21.19$\times$ |
>
> Across all three metrics, AWM reaches the same target quality with substantially lower compute.
>
> We would like to clarify that Gaussian flow matching and diffusion belong to the same generative model family, differing mainly in parameterization, noise schedule, and loss weighting [1][2]. Consequently, Flow-GRPO and DDPO-style methods share the same reverse-step Gaussian likelihood structure, with the main difference lying in the backbone parameterization. Our method and analysis apply to both diffusion and flow-matching models.
>
>
> [1] Yaron Lipman, Ricky T. Q. Chen, Heli Ben-Hamu, Maximilian Nickel, Matt Le. *Flow Matching for Generative Modeling*. ICLR 2023.
>
> [2] Ruiqi Gao, Emiel Hoogeboom, Jonathan Heek, Valentin De Bortoli, Kevin P. Murphy, Tim Salimans. *Diffusion Meets Flow Matching: Two Sides of the Same Coin*. Blog post, `diffusionflow.github.io`.
>
> **Question 2:** The analysis, though, simplifies the basic algorithm by ignoring error terms related to discretisation and bias.
> **Answer 2:** Thank you for this observation. There is perhaps a misunderstanding. Analyzing the idealized setting without discretization error is a deliberate and standard theoretical choice: it establishes a **best-case baseline** for DDPO. In other words, this is not a simplification in our analysis, but rather underscores a **core, inherent flaw of the DDPO algorithm itself**, since DDPO relies on a solver-discretized reverse transition $p_\theta(x_{t-\Delta t}\mid x_t)$. Our theorem analyze an idealized/oracle DDPO without discretization error; even there, the noisy-target issue remains, while discretization bias is an additional solver-dependent limitation of DDPO. AWM, in contrast, is defined on forward-process pairs $(x_0,x_t)$ and avoids reverse-solver discretization.
>
> **Question 3:** The ablations and presentation could be improved.
> **Answer 3:** Thank you for the helpful suggestion. We agree that the presentation can be improved, and we will revise the paper to make it clearer and easier to read. We will also tighten presentation by expanding undefined acronyms, explicitly referencing Figure 2, and reorganizing local exposition.
>
> To further address the concern, we also provide three **additional ablation studies** as show below:
>
> - **Ablation on Monte Carlo steps in Eq. (12).**
>
> We compare $T=6$ (our final setting) and $T=4$, where $T$ is the number of Monte Carlo timesteps used to estimate the DSM-based surrogate in Eq. (12).
> |GenEval: step  | 50 | 100| 200 | 300 | 400| 500 | 600|
> |---|---|---|---|---|---|---|---|
> | AWM with $T=6$ (final version)|0.890| 0.927 | 0.957 | 0.970 |0.975  |0.978  | 0.980|
> | AWM with $T=4$ | 0.860 | 0.901 | 0.928 | 0.950| 0.961 | 0.960 |0.972 |
>
> $T=6$ is slightly more expensive per optimization step than $T=4$, it consistently yields larger reward gains per step and a better final GenEval score. We therefore use $T=6$ in the final version.
>
> - **Ablation study on rollout models $\theta_{\text{old}}$.**
>
> We compare the final setting $\theta_{\text{old}}=\theta$ against using an EMA rollout model $\theta_{\text{old}}=\theta_{\text{EMA}}$ with importance sampling, where the EMA rate is set to $\eta=\min\{0.001t,0.3\}$.
> |GenEval: step  | 50 | 100| 200 | 300 | 400| 500 | 600|
> |---|---|---|---|---|---|---|---|
> | AWM with $\theta_{\text{old}} = \theta$ (final version)|0.890| 0.927 | 0.957 | 0.970 |0.975  |0.978  | 0.980|
> | AWM with $\theta_{\text{old}} = \theta_{\text{EMA}}$ + Importance Sampling | 0.886 |0.947 | 0.963|0.970 |0.974 |0.976 | 0.975|
>
> The two choices perform similarly overall. While the EMA + importance sampling variant is competitive at some intermediate stages, it does not improve the final performance. For simplicity and better compatibility with large-scale industrial training, especially MoE settings (router), we adopt the on-policy choice $\theta_{\text{old}}=\theta$ in the final version.
>
>
> - **Ablation on the KL regularization coefficient $\beta$.**
>
> We compare the final choice $\beta=1$ with a stronger regularization setting $\beta=3$.
> |GenEval: step  | 50 | 100| 200 | 300 | 400| 500 | 600|
> |---|---|---|---|---|---|---|---|
> | AWM with $\beta=1$ (final version)|0.890| 0.927 | 0.957 | 0.970 |0.975  |0.978  | 0.980|
> | AWM with $\beta=3$| 0.880 | 0.931|0.960| 0.954 | 0.956 | 0.967| 0.967|
>
> A larger KL coefficient can be competitive early in training, but it slows later improvement and leads to a lower final GenEval score. This suggests that overly strong regularization hinders optimization, so we use $\beta=1$ in the final version.

---

> > ### Author Rebuttal · Reviewer_a4zM · 2026-04-04
> >
> > I thank the authors for the clarifications and the extra results. I maintain my assessment and scores.

---

> > > ### Author Response · Authors · 2026-04-04
> > >
> > > Thank you for your thoughtful follow-up and for taking the time to consider our rebuttal. We sincerely appreciate your recognition of our clarifications and additional results. Thank you again for your time and feedback throughout the review process.

---

### Decision · Program_Chairs · 2026-04-30

**Decision:**

Accept (regular)

**Comment:**

The paper carries out a theoretical analysis of DDPO and links it to flow matching.  This perspective allows them to propose a new schemed for post-training of diffusion models that aligns with the pre-training objective (the common practice in LLMs). All reviewers acknowledge that the experimental results are promising. In general this is a principled approach and even though some questions were raised about the theoretical details, the authors made a solid effort in their rebuttal.